# ODEBrain: Continuous-Time EEG Graph for Modeling Dynamic Brain Networks

**Haohui Jia**♠, **Zheng Chen**♣,†, **Lingwei Zhu**◇, **Rikuto Kotoge**♣, **Jathurshan Pradeepkumar**♡,
**Yasuko Matsubara**♣, **Jimeng Sun**♡, **Yasushi Sakurai**♣, **Takashi Matsubara**♠
♠ Faculty of Information Science and Technology, Hokkaido University, Japan
♣SANKEN, The University of Osaka, Japan
◇Great Bay University, China
♡Department of Computer Science, University of Illinois Urbana-Champaign, USA
†Corresponding author: `chenz@sanken.osaka-u.ac.jp`

## Abstract

Modeling neural population dynamics is crucial for foundational neuroscientific research and various clinical applications. Conventional latent variable methods typically model continuous brain dynamics through discretizing time with recurrent architecture, which necessarily results in compounded cumulative prediction errors and failure of capturing instantaneous, nonlinear characteristics of EEGs. We propose ODEBRAIN, a Neural ODE latent dynamic forecasting framework to overcome these challenges by integrating spatio-temporal-frequency features into spectral graph nodes, followed by a Neural ODE modeling the continuous latent dynamics. Our design ensures that latent representations can capture stochastic variations of complex brain states at any given time point. Extensive experiments verify that ODEBRAIN can improve significantly over existing methods in forecasting EEG dynamics with enhanced robustness and generalization capabilities. Our code is available at https://github.com/HHJIAnmo/ODEBRAIN.

## 1 Introduction

Modeling dynamic activity in brain networks or connectivity using electroencephalograms (EEGs) is crucial for biomarker discovery (Rolls et al., 2021; Jones et al., 2022) and supports a wide range of clinical applications (Kotoge et al., 2024; Pradeepkumar et al., 2026). Temporal graph networks (TGNs), which integrate temporally sequential models with graph neural networks (GNNs), have recently emerged as a promising approach (Tang et al., 2022; Ho & Armanfard, 2023; Delavari et al., 2024; Li et al., 2024). These methods represent multi-channel EEGs as graphs, where GNNs capture spatial dependencies and sequential models capture fine-grained temporal dynamics, thereby providing mechanistic insights into gradual evolution of brain networks.

However, a critical, yet overlooked problem remains: existing methods transform EEGs into fixed discrete time steps, which conflict with the inherently continuous nature of brain networks. Such discretization imposes rigid windowing assumptions and prevents the models from capturing the unfolding time-course dynamics or irregular transitions in brain networks, as in Figure 1. This paper aims to tackle this issue by developing a novel method that models EEGs in an explicitly continuous manner, while leveraging Neural Ordinary Differential Equations (NODEs) (Chen et al., 2018).

In contrast to sequential models based on recurrent neural networks (RNNs), which discretize time in fixed steps, NODEs parameterize the derivative of the hidden state and integrate it continuously over time (Park et al., 2021). This formulation provides a principled method to model the dynamical evolution of neural activity (Hu et al., 2024) and has been studied across domains (Hwang et al., 2021), including brain imaging (Han et al., 2024). In this paper, we study a novel and critical problem: modeling the dynamics of brain networks with NODEs to learn informative continuous-time representations from EEGs. This remains an unexplored and non-trivial task, and we focus on two main challenges:

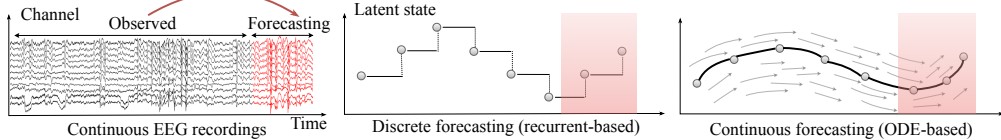

Figure 1: (Left) Continuous EEG real-time neuronal activity recordings. (Mid) Recurrent-based methods employ discrete modeling. (Right) ODE provides a continuous representation for forecasting neuronal population dynamics.

(ⅰ) *Effective spatiotemporal modeling for ODE initialization.* NODEs critically depend on the quality of their initial conditions because the ODE solver propagates trajectories starting from this initialization. Accordingly, a poor initialization propagates errors and destabilizes long-term dynamics. However, EEG signals are inherently noisy and stochastic, making learning robust spatiotemporal representations of brain networks particularly challenging. Therefore, designing an initialization that captures meaningful spatiotemporal structures is essential for stable ODE integration and effective downstream learning.

(ⅱ) *Accurate trajectory modeling.* Trajectory modeling is essential for NODEs, as their strength lies in learning continuous latent dynamics rather than discrete predictions. Unlike conventional time-series data, which often exhibit stable patterns such as periodicity or long-term trends (Klötergens et al., 2025), EEG signals are highly variable, which makes trajectory learning particularly challenging. Thus, a major challenge is to constrain and preserve meaningful trajectories in the latent space so that NODEs can accurately capture the continuous dynamics of EEGs.

In this paper, we introduce a new continuous-time EEG Graph method, ODEBRAIN, based on the NODE, to model dynamic brain networks. To address the above challenges, we first propose a dual-encoder architecture to provide effective initialization for NODEs. One encoder captures deterministic frequency-domain observations to model brain networks, whereas the other integrates raw EEG representations to retain stochastic characteristics. This combination yields robust spatiotemporal features for initializing the ODE solver. Second, we introduce a trajectory forecasting decoder that maps latent representations obtained from NODE solutions back to graph structures. Thereafter, a multistep forecast loss function was applied to explicitly predict future brain networks at different time steps. This design enables direct trajectory modeling of dynamic brain networks and improves accuracy. Third, beyond modeling, we are the first to propose the use of the gradient field of NODEs as a metric to quantify the dynamics of the brain network. Herein, we also conduct a case study on seizure data to illustrate the clinical interpretability of this method.

- **New Problem Formulation.** To the best of our knowledge, we are the first to explicitly formulate EEG brain networks as a continuous-time dynamical system, where the brain network is represented as a sequence of time-varying graphs whose latent dynamics is governed by a NODE. This perspective differs from prior approaches based on recurrent models, which model gradual state transitions in a discrete-time framework rather than in a principled continuous-time manner.

- **Novel Method.** We develop the ODEBRAIN framework that integrates three key components. It first combines deterministic graph-based features with stochastic EEG representations to produce a robust initial state. Then an explicit trajectory forecasting decoder with multi-step forecasting loss hat models temporal–spatial dynamics continuously, enabling principled forecasting of evolving brain networks.

- **Comprehensive Evaluation.** We demonstrate strong performance across benchmarks and provide retrospective clinical case studies that highlight interpretability. Our ODEBRAIN outperforms all baselines in the TUSZ dataset, achieving $6.0\%$ and $8.1\%$ improvements in F1 and ACC, respectively. In the TUAB, ODEBRAIN consistently achieves the best performance, such as $1.2\%$ improved F1 and $2.4\%$ improved AUROC. Moreover, we further evaluate the learned field and its clustering to reveal the dynamic behaviors (varying speed and direction) between the seizure and normal states, and achieve $12.0\%$ improvement for the prediction of brain connectivity.

## 2 RELATED WORKS

### 2.1 TEMPORAL GRAPH METHODS FOR MODELING EEG DYNAMICS

GNNs have emerged as a powerful method for effectively capturing spatial dependencies and relational structures in the analysis of brain networks (Li, 2022; Yang & Hong, 2022; Kan et al., 2023). Specifically, EEG-GNN performs a learnable mask to filter the graph structure of EEG for cognitive classification tasks (Demir et al., 2021). ST-GCN formulates the connectivity of spatio-temporal graphs to capture non-stationary changes (Gadgil et al., 2020). (Tang et al., 2022) have introduced the DCRNN approach for graph modeling, setting a new standard for SOTA in seizure detection and classification tasks. Following this, GRAPHS4MER (Tang et al., 2023) improved the graph structure and integrated it with the MAMBA framework to improve long-term modeling capabilities. AMAG (Li et al., 2024) forecasting method has been proposed to effectively capture the causal relationship between past and future neural activities, demonstrating greater efficiency in modeling dynamics. More recently, EvoBrain has investigated the expressive power of TGNs in integrating temporal and graph-based representations to model brain dynamics (Kotoge et al., 2025). However, these studies rely on discrete modeling and may lead to suboptimal representation of continuous dynamics of brain networks.

### 2.2 DIFFERENTIAL EQUATIONS FOR BRAIN MODELING

Modeling brain function as low-dimensional dynamical systems via differential equations has been a long-standing direction in neuroscience (Churchland et al., 2012; Mante et al., 2013; Vyas et al., 2020), and nonlinear EEG analysis for brain activity mining (Pijn et al., 1997; Xue et al., 2016; Lehnertz et al., 2003; Lehnertz, 2008; Mercier et al., 2024). Recently, NODEs formulate dynamical systems by parameterizing derivatives with neural networks and have shown impressive achievements in various fields (Fang et al., 2021; Hwang et al., 2021; Park et al., 2021). In the modeling of BCI and epilepsy, controllable formulations and fractional dynamics provide important theoretical foundations for modeling brain dynamics (Gupta et al., 2018b; Tzoumas et al., 2018; Lu et al., 2021; Martis et al., 2015; Lepeu et al., 2024). In latent-variable dynamics models, the EEG and neuronal processes are described as fractional dynamics (Gupta et al., 2019; 2018a; Yang et al., 2019; 2025). In neuroscience, (Kim et al., 2021) learn neural activities by modeling the latent evolution of nonlinear single-trial dynamics with Gaussian processes from neural spiking data. (Hu et al., 2024) propose using a smooth 2D Gaussian kernel to represent spikes as latent variables and describe the path dynamics with linear stochastic differentiable equations (SDEs). Another study (Cai et al., 2023) demonstrates robust neuroimaging performance by combining biophysical priors with NODEs, starting from predefined cognitive states. (Chen et al., 2024) have shown the advantage of graph ODE by modeling continuous-time propagation for the EEG emotion task. (Han et al., 2024) further illustrate that integrating spatial structure with NODEs can effectively facilitate the modeling of neuroimaging dynamics, even in the presence of missing data. However, these studies focus on imaging data or neuronal feature engineering, while data-driven modeling of brain networks with fractional dynamics from EEGs remains underexplored.

## 3 PRELIMINARY AND PROBLEM FORMULATION

**Neural Ordinary Differential Equations.** NODEs (Chen et al., 2018) provide a framework for modeling continuous-time dynamics by parameterizing the derivative of a hidden state using neural networks. Intuitively, NODEs solve the trajectory of the hidden state continuously at any arbitrary time $\tau$, rather than restricting updates to fixed discrete steps $\Delta t$ in RNNs. Specifically, the hidden dynamics are computed via an adaptive numerical ODE solver:

$$\boldsymbol{z}(t+1) \simeq \texttt{ODEsolver}(\boldsymbol{z}_0, f_\theta) = \boldsymbol{z}_0 + \int_t^{t+1} f_\theta(t, \boldsymbol{z}_t)\, dt, \qquad (1)$$

where $f_\theta$ is a continuous, differentiable function parameterized by a neural network. This formulation yields a unique continuous trajectory $\boldsymbol{z}(t)$ over an interval $[t_0, t_0 + \tau]$.

**Intuition in Modeling EEG Dynamics.** Conventional sequential models, such as RNNs, have been a standard tool for modeling EEG. However, they implicitly assume that time can be discretized into

fixed steps and that state transitions, such as the onset of a seizure, must occur exactly at these steps (Kotoge et al., 2025). While this assumption simplifies the computation, it poorly matches the reality of EEG, where brain activity evolves continuously and transitions can occur at arbitrary points in time. In contrast, NODEs address this limitation by modeling EEG dynamics through a continuous function $f_\theta$ whose integration yields smooth latent trajectories. Within this framework, discrete EEG signals recorded at sampling intervals are treated as observations sampled from an underlying continuous process $\int f_\theta(t)\, dt$. This perspective allows NODEs to capture both gradual oscillatory rhythms and abrupt transitions in neural activity, thereby providing a more faithful representation of the EEG brain dynamics.

However, applying the NODE to EEG is non-trivial, and we recognize two questions needing to be answered:

1. *Robust initialization $z_0$ against transients and stochasticity in EEGs.* NODE requires a well-cablibrated starting condition $z_0$ to effectively forecast future behavior. This is because EEGs are highly stochastic, or even chaotic to an extent. Their key features are transient and may appear without any preindicator (Chen et al., 2022). Without proper initialization $z_0$ as a guide, the integration of the model $f_\theta$ over time alone cannot accurately forecast future states.

2. *Meaningful objectives of $f_\theta(t, z_t)$ to capture the underlying EEG dynamics.* Standard NODE training typically relies on regression-like objectives aimed at forecasting future states. A key challenge lies in identifying which representations best capture the underlying neural dynamics, so that $f_\theta(t, z_t)$ is guided toward modeling the true evolution of brain networks rather than only surface-level predictions. For example, in seizure analysis, the model must also learn to discern not only seizure but also any leading states that herald an impending seizure (Li et al., 2021).

**Problem Statement (Modeling Dynamic Brain Networks).** Given the observed EEG up to time $t$, denoted as $\mathbf{X}_{\leq t}$, the objective is to model the dynamics of the brain network and forecast their future evolution. The predicted dynamics act as representations of brain states, enabling the distinction between conditions such as seizure and non-seizure. Following prior work (Tang et al., 2022; Chen et al., 2025), we represent the brain as a graph and aim to develop an EEG-based NODE ($\Omega$) to predict a sequence of time-varying graphs:

$$\mathcal{G}_{t+1:t+K} = \{\mathbf{X}_{t+1}, \ldots, \mathbf{X}_{t+K}\} = \Omega\big(z_0, f_{\theta(\mathcal{G}_{1:t})}\big). \tag{2}$$

Over the next $K$ steps graph, where $\mathcal{G}_{1:t}$ denotes the observed brain networks up to time $t$, and $\mathcal{G}_{t+1:t+K}$ represents the predicted dynamic brain networks. These graphs characterize dynamic brain networks, but this problem poses two key challenges: (i) obtaining a robust initialization $\mathbf{z}_0$ that can resist the transient and stochastic nature of EEGs; and (ii) defining an objective for $f_\theta$ that faithfully captures the underlying neural dynamics.

## 4    METHODOLOGY

Figure 2 shows the system overview of ODEBRAIN. Specifically, graph representations are obtained from each EEG segment (Section A.2), entering stage 1: attaining reverse initial state encoding $z^g$ and temporal encoding $z^s$ (Section 4.1). Stage 2 consists of a Neural ODE that takes as input $z^g, z^s$ (Section 4.2). Finally, forecasting loss between ODE output and ground truth is computed.

### 4.1    STAGE 1: REVERSE INITIAL STATE ENCODING

**Spectral Node Encoding.** Previous discrete forecasting studies have shown that the capacity to estimate future neural dynamics depends on past activity in (Li et al., 2024). We also define this forecasting paradigm within our ODEBRAIN. Intuitively, both the latent initial state $z_0$ and the field $f$, i.e., $\frac{dz(t)}{dt}$ are described by encoding the past observation $\mathcal{G}_{i,\leq t}$ to govern the latent continuous evolution. The works of (Rubanova et al., 2019; Chen et al., 2018) suggest that the construction of an effective latent initial state requires an autoregressive model capable of jointly extracting both the initial condition and the latent evolution. Accordingly, we introduce a graph state descriptor $\Phi : \mathbb{R}^d \mapsto \mathbb{R}^m$ that represents latent graph states $z^g \in \mathbb{R}^m$ using the autoregressive and graph network modules.

**Stage 1 Reverse initial state encoding**

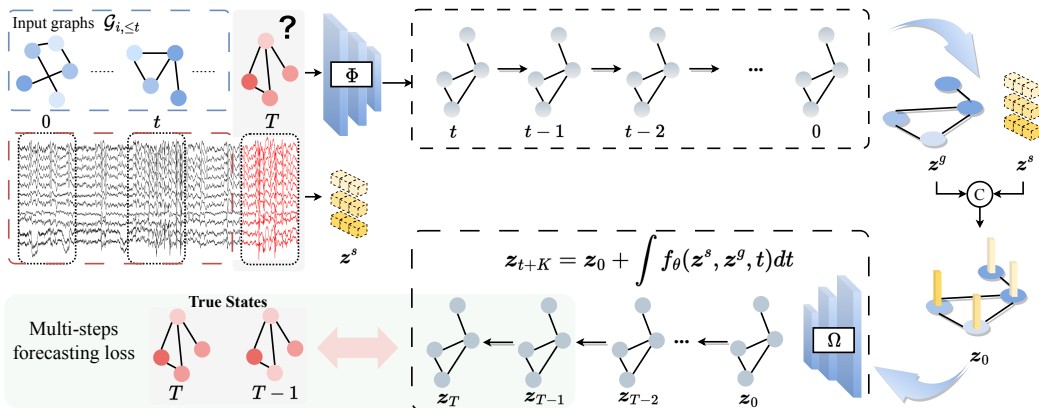

**Stage 2 Forward temporal-spatial ODE Solving**

Figure 2: Continuous neural dynamics modeling via ODEBRAIN with graph forecasting. In stage 1, multi-channel EEG signals are encoded into spectral graph snapshots and fused with raw features to construct noise-robust initial states for ODE integration, predicting future spectral graphs. In stage 2, ODEBRAIN propagates latent states through time, generating dynamic field $f$ that captures continuous trajectories. Finally, future graph node embeddings are obtained by $z_T$ and compared with the ground-truth graph nodes.

Specifically, given the observations until now $\mathcal{G}_{i,\leq t}$ as input, we perform a sequence representation for the node and edge attributes. For node embeddings, the evolution is computed by the gated recurrent unit (GRU) as $\boldsymbol{h}_i^n = \mathrm{GRU}^{\mathrm{node}}(\mathcal{X}_{i,\leq t})$ where $\mathcal{V}_{i,\leq t}$ denotes the spectral attribute sequences of node $i$ and $\mathcal{X}_{i,\leq t}$ the spectral intensity. Similarly, for edge, the attribute sequences are defined from the adjacency matrices by $\boldsymbol{h}_{ij}^e = \mathrm{GRU}^{\mathrm{edge}}(\mathcal{A}_{ij,\leq t})$. The resulting node and edge embeddings are integrated into an aggregated graph structure $\mathcal{G} = (\boldsymbol{h}_{i,t}^n, \boldsymbol{h}_{ij,t}^e)$ to be learned by a GNN to capture the spatial dependency across epochs: $\boldsymbol{z}^g = \mathrm{GNN}(\boldsymbol{h}_i^n, \boldsymbol{h}_{ij}^e)$. The forward process of $\Phi$ captures both the epoch variations between frequency bands and explicit channel correlations.

**Temporal Embedding with Stochasticity.** Accurate modeling of the temporal evolution of EEG signals is crucial because neural dynamics inherently exhibits nonuniform temporal fluctuations and asynchronous activations across channels. Although the graph descriptor $\Phi$ effectively captures the evolution of the node and edge attributes, the short-time Fourier transform (STFT) segments the EEG signals by constant windows, which inevitably disrupts the continuous temporal correlation between the raw EEG observations.

Moreover, fully deterministic latent representations lack the flexibility necessary to effectively represent *transient motions* of EEG as analyzed in Section 3. Conversely, introducing controlled randomness into temporal embeddings serves as a natural regularization strategy, effectively increasing the robustness and preventing premature convergence to suboptimal. Here, we apply the temporal descriptor $\Psi : \mathbb{R}^{T \times L} \mapsto \mathbb{R}^c, c \ll m$ to quantify the randomness of the raw EEG epochs across $N$ channels into $\boldsymbol{z}^s \in \mathbb{R}^c$. Given EEG segments $\mathbf{X}$ from $N$ channels within a sliding window length $L$, we define stochastic temporal embedding as $\boldsymbol{z}^s = \Psi(\mathbf{X}_{T \times L, \leq N})$. The controlled stochasticity further acts as a form of latent space regularization, enhancing generalization and robustness to noise in EEG data collection.

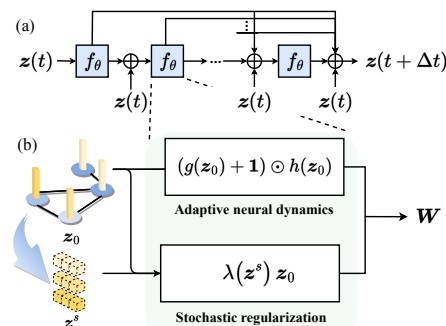

Figure 3: The structure of the temporal-spatial ODE. (a) RK-4 step numerical solver. (b) Procedure of temporal-spatial $f_\theta$.

## 4.2 STAGE 2: FORWARD TEMPORAL-SPATIAL ODE SOLVING

Depending on the above encoding process, we define the initial state $z_0 = [z^s, z^g]$ with $\Phi \circ \Psi \mapsto \mathbb{R}^{m+c}$, which summarizes stochastic temporal variability and deterministic spectral connectivity, respectively. Given the initial state $z_0 \in \mathbb{R}^{m+c}$, the general approach models the ODE vector field following the classical neural network solution $f_\theta$ with residual connection as:

$$d\boldsymbol{z}(t) \equiv f_\theta(\boldsymbol{z}(t), t; \boldsymbol{\Theta})dt, \quad \boldsymbol{z}_0 = [\boldsymbol{z}^s, \boldsymbol{z}^g], \quad t \in [t+1, t+K] \tag{3}$$

where $f_\theta : \mathbb{R}^{m+c} \mapsto \mathbb{R}^{m+c}$ represents a vector field to capture complicated dynamics and its continuous evolution is governed by $f_\theta$ with the learnable $\boldsymbol{\Theta}$ across the entire epoch sequences. However, this introduces the challenge of optimizing the deep network-based $f_\theta$ over highly variable EEG states, making large solver errors.

Considering the deep architecture-based multi-step numerical solver design (Lu et al., 2018; Oh et al., 2024) and the logic gating interaction of brain dynamics (Goldental et al., 2014), we design a temporal-spatial ODE solution to incorporate the initial state $z_0$ for additive and gate operations as shown in Figure 3. In addition, we further introduce an adaptive decay component conditioned on the stochastic temporal state $z^s$, to adjust the vector field $f_\theta$, accounting for the complexity and dynamic nature of the brain as a system. As shown in Figure 3(b), the $f_\theta$ used in the proposed ODE function is defined as follows:

$$f_\theta(\boldsymbol{z}_0) = (g(\boldsymbol{z}_0) + \boldsymbol{1}) \odot h(\boldsymbol{z}_0) - \lambda(\boldsymbol{z}^s)\,\boldsymbol{z}_0, \quad \boldsymbol{z}_0 = [\boldsymbol{z}^s, \boldsymbol{z}^g], \tag{4}$$

where $\odot$ represents the element-wise multiplication. Initially, the vector field is computed by the general residual block $h(\boldsymbol{z}_0)$ and updated by a gated vector field with a sigmoid function $\sigma$ as:

$$g(\boldsymbol{z}_0) = \sigma(W_g \boldsymbol{z}_0 + \boldsymbol{b}_g) \in (0, 1)^{m+c}, \tag{5}$$

which provides state-adaptive modulation of the dynamics. Finally, to regularize trajectories under noisy EEG inputs, we add an adaptive decay conditioned on the temporal stochastic state $z^s$:

$$\lambda(\boldsymbol{z}^s) = \texttt{Softplus}(W_a^{(2)} \circ \texttt{tanh}(W_s^{(1)}\boldsymbol{z}^s + \boldsymbol{b}^1) + \boldsymbol{b}^2) > 0. \tag{6}$$

The latent trajectory $\boldsymbol{z}(t)$ at an arbitrary time $t$ can be solved by:

$$\boldsymbol{z}_{t+K} = \begin{bmatrix} \boldsymbol{z}^s \\ \boldsymbol{z}^g \end{bmatrix} + \int_{t+1}^{t+K} f_\theta\left(\begin{bmatrix} \boldsymbol{z}^s \\ \boldsymbol{z}_t^g \end{bmatrix}, t\right) dt \quad . \tag{7}$$

The state solutions are calculated by solving with efficient numerical solvers in Figure 3(a), such as Runge-Kutta (RK) (Schober et al., 2019). The latent state at the next timestamp is updated as follows:

$$\boldsymbol{z}(t + \Delta t) = \boldsymbol{z}(t) + \frac{\Delta t}{6}(k_1 + 2k_2 + 2k_3 + k_4). \tag{8}$$

## 4.3 GRAPH EMBEDDING FORECASTING

Depending on Equation 7, the latent dynamic function and neural forecasting are presented as follows:

$$\{\boldsymbol{z}_{t+1}, \ldots, \boldsymbol{z}_{t+K}\} = \texttt{ODESolver}\left(f_\theta, [\boldsymbol{z}^s, \boldsymbol{z}^g], [t+1, t+K]\right), \tag{9}$$

$$\hat{\mathcal{G}}_{t+i} = \Omega(\boldsymbol{z}_{t+i}) \quad \forall i \in \{1, 2, \ldots, K\}, \tag{10}$$

where continuous latent trajectories $\{\boldsymbol{z}(t)\}_{t=1}^K$ are projected back to the future EEG node attributes with $\mathcal{V}$ the set of all possible unique nodes in $\mathcal{G}_{t+1:t+K}$ via a predictive module $\Omega : \mathbb{R}^{m+c} \mapsto \mathbb{R}^d$, explicitly capturing spatial correlations across the EEG channels over future $K$ time steps. Here, $\mathcal{X}_{:, >t} = [\mathcal{X}_{:,t+1}, \ldots, \mathcal{X}_{:,t+K}]$ integrate all future node attributes.

Unlike the previous works, which focus on forecasting the temporal neural population dynamics. Our learning objective is to predict the graph structure rather than the simple temporal dynamics, since neuron firing generally activates in the asynchronous channels simultaneously $\mathcal{L}_\mathcal{G} = \mathbb{E}_\mathcal{G} \left\| \hat{\mathcal{G}}_{t+1:K} - \mathcal{G}_{t+1:K} \right\|_2$. We first train the model in an unsupervised manner using dynamic graph forecasting loss to capture continuous neural dynamics via ODE solvers. Then we pool the latent continuous trajectory $z(t)$ extracted from the ODE solver with entire timesteps for downstream fine-tuning, such as classification.

Table 2: Main results on TUSZ (12s seizure detection) and TUAB. **Bold** and underline indicate best and second-best results. $\star$: The performance depends on the discrete multi-steps forecasting. $\dagger$: The performance depends on the *continuous* multi-steps forecasting. $\ddagger$: The performance depends on the *continuous* single-step forecasting.

| Method | TUSZ | | | TUAB | | |
|---|---|---|---|---|---|---|
| | Acc | F1 | AUROC | Acc | F1 | AUROC |
| CNN-LSTM | $0.735 \pm 0.003$ | $0.347 \pm 0.012$ | $0.757 \pm 0.003$ | $0.741 \pm 0.002$ | $0.736 \pm 0.007$ | $0.813 \pm 0.003$ |
| BIOT | $0.702 \pm 0.003$ | $0.294 \pm 0.006$ | $0.772 \pm 0.006$ | $0.717 \pm 0.002$ | $0.713 \pm 0.004$ | $0.788 \pm 0.002$ |
| EvolveGCN | $0.769 \pm 0.002$ | $0.385 \pm 0.005$ | $0.791 \pm 0.004$ | $0.708 \pm 0.003$ | $0.707 \pm 0.002$ | $0.777 \pm 0.003$ |
| DCRNN | $0.816 \pm 0.002$ | $0.416 \pm 0.009$ | $0.825 \pm 0.002$ | $0.768 \pm 0.004$ | $0.769 \pm 0.002$ | $0.848 \pm 0.002$ |
| latent-ODE | $0.827 \pm 0.004$ | $0.470 \pm 0.005$ | $0.849 \pm 0.004$ | $0.749 \pm 0.002$ | $0.745 \pm 0.002$ | $0.829 \pm 0.004$ |
| latent-ODE (RK4) | $0.821 \pm 0.003$ | $0.465 \pm 0.001$ | $0.845 \pm 0.004$ | $0.746 \pm 0.002$ | $0.739 \pm 0.002$ | $0.823 \pm 0.003$ |
| ODE-RNN | $0.802 \pm 0.002$ | $0.455 \pm 0.007$ | $0.855 \pm 0.003$ | $0.751 \pm 0.003$ | $0.744 \pm 0.004$ | $0.838 \pm 0.005$ |
| neural SDE | $0.857 \pm 0.002$ | $0.467 \pm 0.003$ | $0.851 \pm 0.002$ | $0.768 \pm 0.003$ | $0.751 \pm 0.003$ | $0.834 \pm 0.002$ |
| GDEs | $0.849 \pm 0.003$ | $0.475 \pm 0.005$ | $0.841 \pm 0.003$ | $0.757 \pm 0.003$ | $0.737 \pm 0.006$ | $0.823 \pm 0.004$ |
| ODEBRAIN$^\dagger$ | $0.869 \pm 0.003$ | $0.488 \pm 0.015$ | $0.875 \pm 0.005$ | $0.771 \pm 0.005$ | $0.770 \pm 0.005$ | $0.849 \pm 0.003$ |
| ODEBRAIN$^\ddagger$ | $\mathbf{0.877 \pm 0.004}$ | $\mathbf{0.496 \pm 0.017}$ | $\mathbf{0.881 \pm 0.006}$ | $\mathbf{0.778 \pm 0.003}$ | $\mathbf{0.774 \pm 0.005}$ | $\mathbf{0.857 \pm 0.005}$ |

## 5 EXPERIMENTS

In this section, we conduct experiments to answer the following research questions:
- RQ1. Does ODEBRAIN strengthen seizure detection capability through continuous forecasting?
- RQ2. How does the initial state $z_0$ affect the development of the latent neural trajectory?
- RQ3. Does the objective of $\Omega$ facilitate dynamic optimization?
More detailed experiment settings can be found in the Appendix A.

### 5.1 EXPERIMENTAL SETUP

**Tasks.** In this study, we evaluate our ODEBRAIN for modeling neuronal population dynamics with seizure detection and abnormal EEG classification. We select these tasks because they reflect rapid, highly non-stationary transitions and clinically abnormal neural patterns. We provide detailed information about the datasets in Section A.3.

**Baseline methods.** We select two studies of dynamic neural population: EvolveGCN and DCRNN (Li et al., 2017; Pareja et al., 2020) with a reconstruction objective. We also compare it with the benchmark Transformer BIOT (Yang et al., 2023), which captures temporal-spatial information for EEG tasks, and a standard CNN-LSTM (Ahmedt-Aristizabal et al., 2020). For continuous modeling methods, we select the normal NODE (Chen et al., 2018), an enhanced ODE-RNN for irregular time modeling with autoregressive vector field (Rubanova et al., 2019), stochastic-regularized neural SDE (Liu et al., 2019) and graph-defined GDEs (Poli et al., 2019).

**Metrics.** To answer **RQ1**, we evaluate the model using the Area Under the Receiver Operating Characteristic Curve (AUROC) and the F1 score. The AUROC measures the ability of the models across varying thresholds, while the F1 score highlights the balance between precision and recall at its optimal threshold for classification. For **RQ2**, we measure the structural similarity of the predicted graph using the

Table 1: Results (AUROC↑, F1↑) on TUSZ (12s and 60s seizure detection) against discrete and continuous baselines, with options on the gate and stochastic regularization. (-: w/o, +Random: gate with random coefficients for stochastic regularization.) Bold = best.

| Model | Method | T(s) | AUROC | F1 |
|---|---|---|---|---|
| Discrete & Continuous | BIOT | 12 | $0.772 \pm 0.006$ | $0.294 \pm 0.006$ |
| | | 60 | $0.642 \pm 0.009$ | $0.256 \pm 0.003$ |
| | DCRNN | 12 | $0.816 \pm 0.002$ | $0.416 \pm 0.009$ |
| | | 60 | $0.802 \pm 0.003$ | $0.413 \pm 0.005$ |
| | latent-ODE | 12 | $0.791 \pm 0.004$ | $0.385 \pm 0.005$ |
| | | 60 | $0.745 \pm 0.036$ | $0.331 \pm 0.031$ |
| | ODEBRAIN | 12 | $\mathbf{0.881 \pm 0.006}$ | $\mathbf{0.496 \pm 0.017}$ |
| | | 60 | $\mathbf{0.828 \pm 0.003}$ | $\mathbf{0.430 \pm 0.021}$ |
| ODEBRAIN | - Gate | 12 | $0.867 \pm 0.004$ | $0.488 \pm 0.007$ |
| | | 60 | $0.821 \pm 0.034$ | $0.424 \pm 0.003$ |
| | - Stochastic | 12 | $0.848 \pm 0.017$ | $0.462 \pm 0.013$ |
| | | 60 | $0.817 \pm 0.029$ | $0.414 \pm 0.047$ |
| | +Random | 12 | $0.860 \pm 0.017$ | $0.474 \pm 0.033$ |
| | | 60 | $0.819 \pm 0.026$ | $0.418 \pm 0.017$ |

Global Jaccard Index (GJI) $\texttt{GJI}(\mathcal{E}_{true}, \mathcal{E}_{Pred}) = \frac{|\mathcal{E}_{true} \cap \mathcal{E}_{Pred}|}{|\mathcal{E}_{true} \cup \mathcal{E}_{Pred}|}$ (Castrillo et al., 2018). For **RQ3**, We compute the cosine similarity of predicted node embeddings.

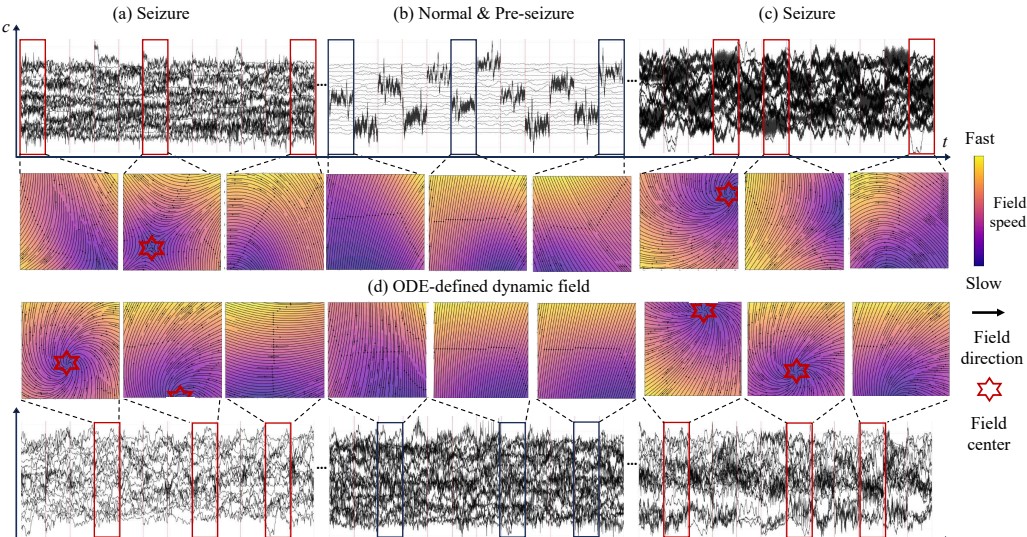

Figure 4: Visualization results between the multichannel EEG signal (upper and lower) and its latent dynamic field $f_\theta$ (middle) obtained by ODEBRAIN. Local minima appearing in (a) and (c) indicate rapid changes, corresponding to seizure states.

## 5.2 RESULTS

### 5.2.1 MAIN RESULT

**RQ1** concerns the continuous forecasting capability on EEG. Table 2 summarizes seizure detection accuracy across models on the TUSZ and TUAB datasets for a duration of 12 seconds. Our ODEBRAIN consistently outperforms all baselines based on the AUROC and F1 score, demonstrating the superiorty of continuous forecasting. Notably, our single-step forecasting achieves an AUROC of $0.881 \pm 0.006$ and an F1 score of $0.496 \pm 0.017$, surpassing latent-ODE. Our multi-step forecasting attains a recall of $0.563 \pm 0.015$, balancing overall detection capability and positive-instance coverage. These results indicate that ODEBRAIN is more effective in capturing the transient dynamics of EEGs than the fixed-time interval or reconstruction baselines.

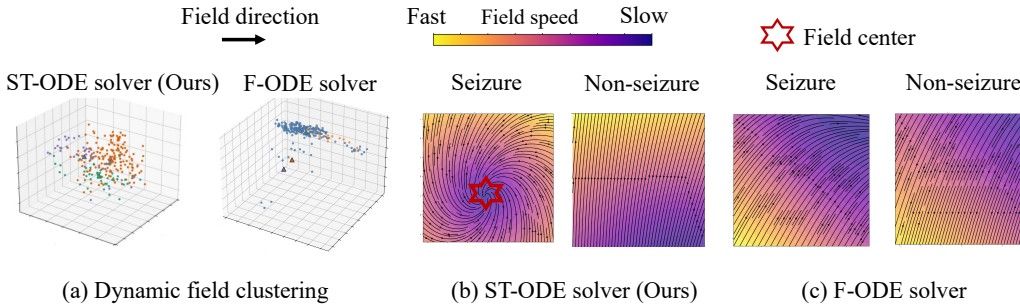

Figure 5: Visualizing learned dynamic fields between our spatial-temporal(ST)-ODE solver and the frequency (F)-ODE solver.

To further illustrate this point, we visualize the dynamic field $f_\theta$ of the latent space in Figure 4. This dynamic field characterizes the difference between seizure and normal states. This is most apparent in the centers in the seizure in Figure 4(a) and 4(c) while absent from the normal & pre-seizure states in Figure 4(b). These centers depict an area where gradients point and eventually the flows converge. This aligns well with the corresponding EEGs that show wild-type oscillations with high frequency components. In contrast, for the normal & pre-seizure data, such centers are not present in the field, showing that the dynamics is driven mainly by low-frequency oscillations. It is worth noting that such visualization is only available to continuous dynamics modeling of our method.

In summary, we can answer **RQ1** as follows: through continuous forecasting, ODEBRAIN outperforms existing baselines in terms of seizure detection by accurately depicting neural population dynamics. The learned field $f_\theta$ can clearly delineate the boundary between seizure and normal states via its vector field representation of neuronal activity. Unlike discrete-time interval and reconstruction-based baselines, ODEBRAIN provides an arbitrary temporal resolution and it is sensitive to transient neural changes. We have verified that it helps capture the transition process of different brain states.

### 5.2.2 DYNAMIC GRAPH FORECASTING EVALUATION

**RQ2** concerns the initial state $z_0$. Figure 5 depicts the learned latent dynamic fields induced by different initialization strategies. The proposed temporal-spatial initialization (ST-ODE solver) yields a markedly more structured latent field than the frequency-based initialization (F-ODE solver). In particular, the ST-ODE field exhibits clearer clustering behavior and more distinguishable dynamic behaviors between seizure and non-seizure states. ODEBRAIN can preserve the transient dynamics and guide the trajectory toward a meaningful dynamical region. We answer **RQ2**, given that our $z_0$, ODEBRAIN can generate more remarkable vector fields that respect the dynamics of EEG and maintain continuous evolutionary properties.

Table 1 describes the performance of seizure detection under 12s and 60s, comparing discrete and continuous baselines with ODEBRAIN. ODEBRAIN achieves the best or tied-best results at both horizons, indicating that the adaptive vector field effectively strengthens stability. The ablations further validate our design by removing the gating mechanism leading to performance drop from $0.881$ to $0.867$, highlighting the adaptive vector field can achieve stable trajectory evolution. Removing stochastic regularization also degrades F1 from $0.496$ to $0.462$, proving that stochastic regularization mitigates the instability caused by noise. In contrast, using a gate with random coefficients for stochastic regularization still underperforms the full model, implying that our learnable regularization is more effective.

**RQ3** concerns consistency in the graphs. Figure 6 shows the effectiveness of our objective $\Omega$ that helps predict dynamic graph structures. It is visible that ODEBRAIN achieves higher similarity scores ($0.53 \rightarrow 0.63$) than the discrete predictor, indicating that ODEBRAIN more accurately captures the true graph structure with the help of $\Omega$. The similarity matrices reveal that ours aligns more closely in terms of local correlation distribution, in which the discrete predictor exhibits notable discrepancies in certain block structures. Now we can answer **RQ3**: the explicit graph embedding target improves forecasting accuracy. This is achieved by guiding the vector field $f_\theta$ to learn continuous trajectories that align well with neural activity, leading to more reliable prediction.

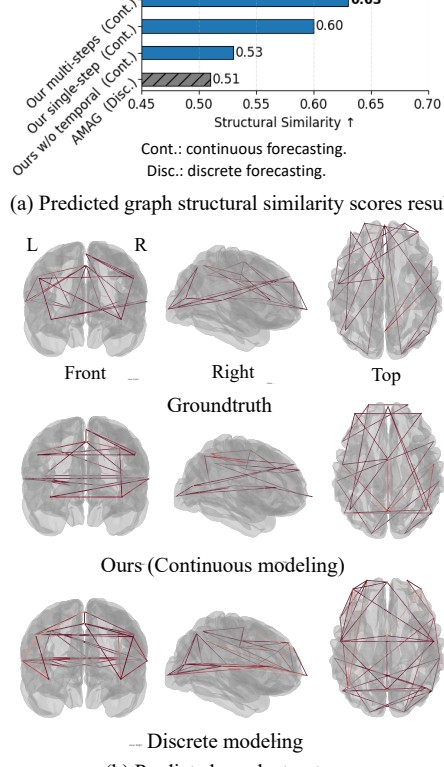

(a) Predicted graph structural similarity scores result

(b) Predicted graph structures

Figure 6: Results on (a) graph similarity and (b) functional connections.

### 5.2.3 ABLATION STUDY

We perform ablation study on the following factors of ODEBRAIN: initialization $z_0$, loss objective $\Omega$ and forecasting horizon, the results are summarized in Figure 7.

**Initial state.** Temporal–spatial initial state option yields the best performance, achieving the highest AUROC ($0.877$) and surpassing spatial-only ($0.862$) and mix up ($0.851$). It mitigates sensitivity to initial conditions and delivers the largest gains at the longest horizon (11s). **Loss objective.** Our structural multi-step forecasting consistently outperforms reconstruction-only and raw-signal

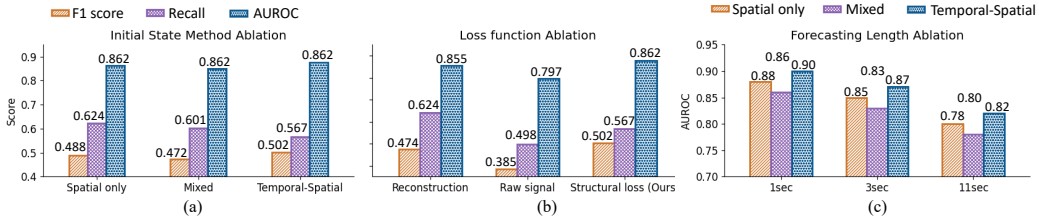

Figure 7: Summary of ablation study. (a) State initialization. We compare spatial-only, mixed, and temporal–spatial initialization and summarized results in F1, Recall and AUROC. Temporal–Spatial achieves the best F1 ($0.502$) with a competitive recall. (b) Loss function. Replacing our structural forecasting loss with reconstruction-only or raw-signal forecasting degrades performance on AUROC. (c) Forecast horizon. AUROC decreases as the horizon grows ($1s \rightarrow 3s \rightarrow 11s$), and Temporal–Spatial remains the best across all horizons over others.

Table 3: Computational cost with wall-clock time (s) and NFEs.

| Type | Model | Param. | Wall | NFEs |
|---|---|---|---|---|
| Discrete | CNN-LSTM | 5976K | 0.586±0.004 | - |
| | BIOT | 3174K | 0.508±0.003 | - |
| | DCRNN | 281K | 0.418±0.006 | - |
| Continuous | latent-ODE | 386K | 0.421±0.002 | 102 |
| | ODE-RNN | 675K | 0.601±0.005 | 189 |
| | neural SDE | 346K | 0.482±0.003 | 153 |
| | `ODEBRAIN` | 459K | 0.516±0.002 | 164 |

Table 4: Ablation study on Top-$\tau$=3 and different regularizer options. Bold denotes the best.

| Model | Regularizer | AUROC | Recall |
|---|---|---|---|
| latent-ODE | Shrinkage | 0.833±0.032 | 0.567±0.021 |
| | Graphical lasso | 0.846±0.025 | 0.557±0.022 |
| | Norm | 0.849±0.004 | 0.575±0.005 |
| ODEBRAIN | Shrinkage | 0.872±0.023 | 0.606±0.035 |
| | Graphical lasso | 0.872±0.017 | **0.613±0.033** |
| | Norm | **0.881±0.006** | 0.605±0.003 |

forecasting across F1/Recall/AUROC, indicating that geometry-aware regularization improves dynamical modeling. We attribute the gains to `ODEBRAIN` that couples the spectral–spatial structure with EEG dynamics and enables more stable integration and stronger generalization.

Table 3 shows single-batch inference cost for discrete vs. continuous baselines, including parameters, wall-clock time, and NFEs (only for solver-based models). Discrete methods have fixed-depth computation, so latency mainly follows model size/sequence length. NFEs are shown only for the ODE solver-based models. `ODEBRAIN` contains 459k parameters with 164 NFEs, and 0.516s per batch, which falls in the same latency band as discrete models with fixed-depth computation. These results indicate that `ODEBRAIN` does not introduce prohibitive cost in practice, and the reduced NFEs suggest a more stable integration than other complicated continuous baselines.

Table 4 evaluates sensitivity to top-$\tau$=3 sparsity and regularization options. Adding regularization improves Recall, confirming that norm correlation graphs are noisy and susceptible to volume conduction, while regularized connectivity is more reliable. The performance is stable across regularization options. Concretely, an ODE solver can achieve better performance with sparser, regularized graphs. Graphical lasso or Norm with 3 sparsity yields the best in both AUROC and Recall. For `ODEBRAIN`, Norm with 3 sparsity achieves the best AUROC (0.881), and Graphical lasso gets the highest Recall (0.613), demonstrating robust dependence on graph-construction choices.

## 6 CONCLUSION

In this work, we introduced `ODEBRAIN`, a novel continuous-time dynamic modeling framework for modeling EEGs, designed explicitly to overcome critical limitations associated with discrete-time recurrent approaches. By adopting a neural ODE-based approach with adaptive vector field strategy, our model effectively captures the continuous neural dynamics and spatial interactions in EEG data. Although `ODEBRAIN` models latent dynamics in continuous time, the inputs and supervision are still based on epoched segments, which limits long-term continuous modeling. The generalization to other neurological disorders or cognitive tasks remains to be explored.

ACKNOWLEDGMENT

This study was partly supported by JST PRESTO (JPMJPR24TB), CREST (JPMJCR24Q5, JP-MJCR23M3, JPMJCR20C6), ASPIRE (JPMJAP2329), and Moonshot R&D (JPMJMS2033-14). JSPS KAKENHI Grant-in-Aid for Scientific Research Number JP24K20778, JST START JP-MJST2553, JST K Program JPMJKP25Y6, JST COI-NEXT (JPMJPF2009, JPMJPF2115), as well as the Future Social Value Co-Creation Project (Osaka University).

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

# A    EXPERIMENTAL SETTINGS

## A.1    DISCUSSION: KEY INSIGHTS OF ODEBRAIN

Conceptually, the main gain of our work comes from explicitly modeling continuous dynamics over graph structures. By capturing the dynamic evolution of EEG signals, the model can effectively handle substantial noise, randomness, and fluctuations. Our comparison with the baseline without continuous dynamics (i.e., using only a temporal GNN backbone) clearly supports this observation. Methodologically, our improvements arise from two key aspects: (i) obtaining a high-quality initialization $\mathbf{z}_0$, and (ii) formulating a vector field $f_\theta$ that captures informative and stable dynamics. First, reverse initial encoding provides a high-quality continuous representation that enables the model to unfold temporal information embedded in EEGs. This is achieved through a dual-encoder architecture that integrates spectral graph features with stochastic temporal signals. Second, the temporal–spatial ODE solver $f_\theta$ incorporates initialization into additive and gating operations, enabling adaptive emphasis on informative EEG connectivity patterns that encode richer dynamics (Figure 4). Furthermore, the stochastic regularizer mitigates the classical error-accumulation problem of ODEs by modeling stochasticity in the EEG time domain, thereby improving long-term stability. We also include a new ablation table (Table 1) to validate the contribution of each component and support the above points.

## A.2    DYNAMIC SPECTRAL GRAPH STRUCTURE

Raw EEG signals consist of complicated neural activities that overlap in multiple frequency bands, each potentially encoding different functional neural dynamics. Directly analyzing EEG signals in the time domain often misses subtle state transitions that occur uniquely within specific frequency bands (Yang & Hong, 2022; Chen et al., 2023). Hence, it is beneficial to represent the intensity variations of frequency bands and waveforms by decomposing raw EEG signals into frequency components. To effectively provide detailed insights for subtle state transitions, we perform the short-time Fourier transform (STFT) to each EEG epoch, preserving their non-negative log-spectral. Consequently, the multi-channel EEG recordings are processed as:

$$\mathbf{X}_t = \sum_{t=\infty}^{-\infty} x[t]\, \omega[t-m] e^{-jwt}, \tag{11}$$

and a sequence of EEG epochs with their spectral representation is formulated as $\mathbf{X} \in \mathbb{R}^{N \times d \times T}$.

We then apply a graph representation by measuring the similarity between the spectral representation $\mathbf{X}$ across the EEG channels. Specifically, we define an adjacency matrix $\mathcal{A}_t(i,j)$ at each epoch $t$ as follows: $\mathcal{A}_t(i,j) = \text{sim}(\mathbf{X}_{i,t}, \mathbf{X}_{j,t})$ and compute the normalized correlation between nodes $v_i$ and $v_j$, where the structure of the graph and its associated edge weight matrix $A_{i,j}$ are inferred from $X_t$ for each $t$-th epoch. We only preserve the highest top-$\tau$ correlations to construct the evident graphs without redundancy. To avoid redundant connections and clearly represent dominant spatial structures, we retain only the top-$\tau$ strongest connections at each epoch for sparse and meaningful graph representations. Thus, we obtain a temporal sequence of EEG spectral graphs $\{G_t = (\mathcal{V}_t, \mathcal{A}_t)\}_{t=0}^T$.

**Temporal Graph Representation.** Taking an EEG $\mathbf{X}$ consisting of $N$ channels and $T$ time points, we represent $\mathbf{X}$ as a graph, denoted $\mathcal{G} = \{\mathcal{V}, \mathcal{A}, \mathbf{X}\}$, where $\mathcal{V} = \{v_1, \ldots, v_N\}$ represents the set of nodes. Each node corresponds to an EEG channel. The adjacency matrix $\mathcal{A} \in \mathbb{R}^{N \times N \times T}$ encodes the connectivity between these nodes over time, with each element $a_{i,j,t}$ indicating the strength of connectivity between nodes $v_i$ and $v_j$ at the time point $t$. Here, we redefine $T$ as a sequence of EEG segments, termed epochs, obtained using a moving window approach. The embedding of node $v_i$ at the $t$-th epoch is represented as $h_{i,t} \in \mathbb{R}^m$. Specifically, we perform the short-time Fourier transform (STFT) on each EEG epoch, referring to (Tang et al., 2022). Then we measure the similarity among the spectral representation of the EEG channels to initial the $\mathcal{A}_t(i,j)$ for each epoch $t$.

## A.3    DATASETS AND EVALUATION PROTOCOLS

**Datasets.** We use Temple University Hospital EEG Seizure (TUSZ) and the TUH Abnormal EEG Corpus (TUAB) (Shah et al., 2018), the largest publicly available EEG seizure database. TUSZ contains 5,612 EEG recordings with 3,050 annotated seizures. Each recording consists of 19 EEG

channels following the 10-20 system, ensuring clinical relevance. A key strength of TUSZ lies in its diversity, as the dataset includes data collected over different time periods, using various equipment, and covering a wide age range of subjects. To provide normal controls, we sample studies from the normal subset of TUAB. Unless stated otherwise, recordings are processed with the same pipeline across corpora (canonical 10–20 montage with 19 channels and unified resampling), ensuring consistent preprocessing for cross-dataset evaluation.

**Metrics.** To answer **RQ1**, we evaluate the model using the Area Under the Receiver Operating Characteristic Curve (AUROC) and the F1 score. AUROC measures the ability of models across varying thresholds, while the F1 score highlights the balance between precision and recall at its optimal threshold for classification. For **RQ2**, we measure the predicted graph structural similarity using the Global Jaccard Index (GJI) (Castrillo et al., 2018):

$$\text{GJI}(\mathcal{E}_{true}, \mathcal{E}_{Pred}) = \frac{|\mathcal{E}_{true} \cap \mathcal{E}_{Pred}|}{|\mathcal{E}_{true} \cup \mathcal{E}_{Pred}|} \quad . \tag{12}$$

**Model training.** All models are optimized using the Adam optimizer (Kingma, 2014) with an initial learning rate of $1 \times 10^{-3}$ in the PyTorch and PyTorch Geometric libraries on NVIDIA A6000 GPU and AMD EPYC 7302 CPU. We adopt the adaptive Runge-Kutta NODE integration solver (RK45) with relative tolerance set to $1 \times 10^{-5}$ for training.

## A.4 HYPERPARAMETERS

All experiments are conducted on the TUSZ and TUAB dataset using CUDA devices and a fixed random seed of 123. EEG signals are preprocessed via the Fourier transform, segmented into 12-second sequences with a 1-second step size, and represented as dynamic graphs comprising 19 nodes (EEG channels). Graph sparsification is achieved with Top-$\tau = 3$ neighbors. Both dynamic and individual graphs use dual random-walk filters, whereas the combined graph employs a Laplacian filter. The default backbone is GRU-GCN for reverse initial state encoding, consisting of 2-layer GRU with 64 hidden units per layer. We also apply a CNN encoder with 3 hidden layers to extract the stochastic feature $z^s$ to obtain the final initial value $z_0$. The convolution adopts a $2 \times 2$ kernel size with batch normalization and max pooling . The input and output feature dimensions are both 100, with the number of classes set to 1 for detection/classification tasks.

We train models using an initial learning rate of 3e-4, weight decay 5e-4, dropout rate 0.0, batch sizes of 128 (training) and 256 (validation/test), and a maximum of 100 epochs. Gradient clipping with a maximum norm of 5.0 and early stopping with patience of 5 epochs are applied. The model checkpoints are selected by maximizing AUROC in the validation set (weighted averaging). When the metric is loss, we instead minimize it; all other metrics (e.g., F1, ACC) are maximized. Data augmentation is enabled by default, while curriculum learning is disabled unless otherwise stated.

## B ADDITIONAL RESULTS

Fig. 8 shows the visualization of the dynamic field $f_\theta$ of the latent space. It reveals distinct neural activity patterns: during synchronous low-frequency oscillations, dynamic field appears steady state, while high-frequency bursts trigger localized positive gradients, driving system activation. Asynchronous cross-channel interactions manifest as vortex-like flows, reflecting dynamic balance. Notably, continuous dynamic evolution offers finer temporal resolution at an arbitrate time. ODEBRAIN enables early detection of neural transitions, better than discrete-time methods.

Fig. 9(a) depicts the predicted connectivity patterns and edge densities of ODEBRAIN closer to the true connectivity than the discrete predictor-based AMAG, leading to a significant topology consistency. These structural features are crucial for modeling consistent brain dynamics, as small topological offsets lead to correct brain activity for downstream tasks. The stochastic components of the raw EEG signal can be regard as an implicit regularity term, which helps to enhance the generalization ability of continuous trajectory inference and maintains consistency with the structure. The latent variable trajectories generated by ODEBRAIN not only maintain continuous evolutionary properties, but also enhance the predictive ability of spatial consistency.

Fig. 9 shows the effectiveness of predicting the dynamic graph structure depending on our meaningful forecasting objective $\Omega$. Fig. 9(b) shows that ODEBRAIN can achieve higher similarity than the

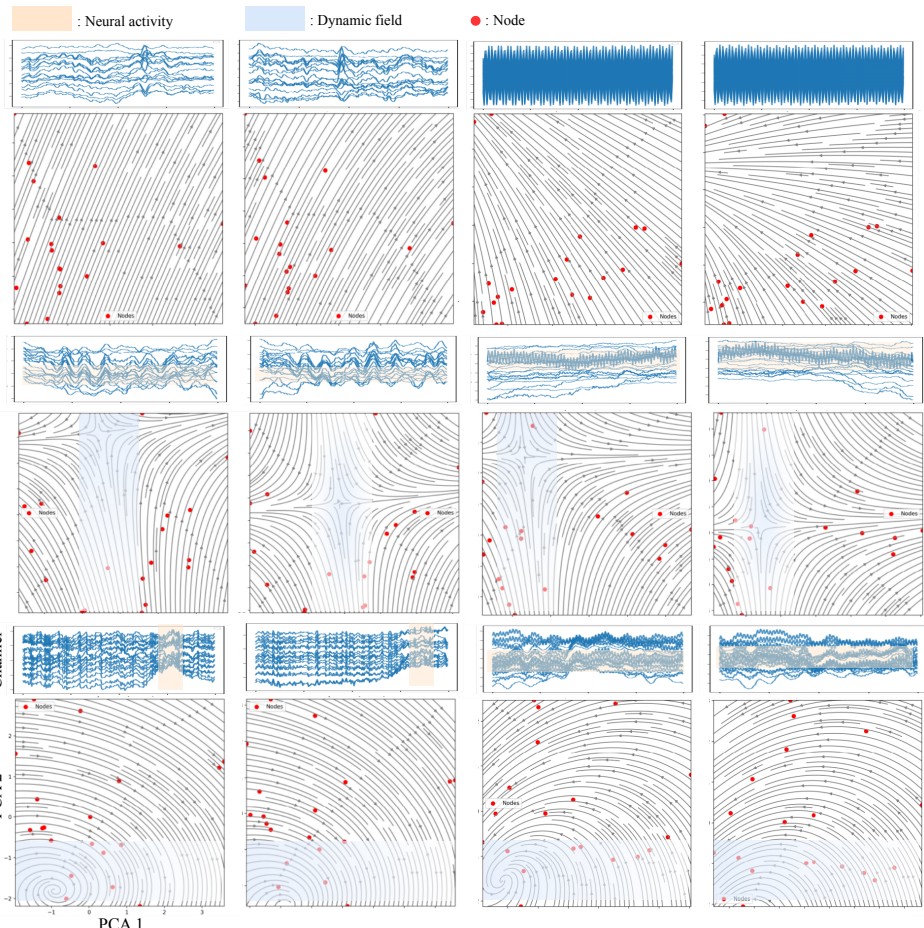

Figure 8: Visualization results between the multichannel EEG signal (upper) and its latent dynamic field $f_\theta$ (lower) in our temporal-spatial neural ODE.

Table 5: Ablation of pooling options over ODE-trajectory on **TUSZ** (12s seizure detection) and **TUAB**. **Bold** indicates best result.

| Method | TUSZ | | | TUAB | | |
|---|---|---|---|---|---|---|
| | Acc | F1 | AUROC | Acc | F1 | AUROC |
| Max pooling | **0.877 ± 0.004** | **0.496 ± 0.017** | **0.881 ± 0.006** | **0.778 ± 0.003** | **0.774 ± 0.005** | **0.857 ± 0.005** |
| Mean pooling | 0.842 ± 0.002 | 0.385 ± 0.005 | 0.827 ± 0.003 | 0.748 ± 0.002 | 0.635 ± 0.002 | 0.827 ± 0.004 |
| Sum pooling | 0.851 ± 0.002 | 0.466 ± 0.005 | 0.867 ± 0.004 | 0.753 ± 0.003 | 0.755 ± 0.002 | 0.831 ± 0.004 |

discrete predictor, indicating that the continuous prediction model more accurately captures the true graph structure. The similarity matrices reveal that ours aligns more closely in terms of local correlation distribution, in which the discrete predictor exhibits notable discrepancies in certain block structures. The explicit graph embedding target improves the forecasting accuracy, while effectively guides the vector field $f_\theta$ to learn continuous trajectories aligned with neural activity, leading to a more reliable prediction.

Table 5 concerns the sensitivity with Top-$\tau$ options ($\tau$=3,7) and different graph regularizers, evaluated under both latent-ODE and ODEBRAIN. Overall, regularized graph construction consistently improves both metrics for the two frameworks, indicating that raw correlation graphs can be vulnerable to noise and volume conduction, while statistical regularization yields more reliable functional connectivity. Specifically, for latent-ODE, graphical lasso and norm regularization with $\tau$=3 achieve the strongest AUROC/Recall, suggesting that a sparser, regularized partial-correlation structure is preferable for continuous dynamics modeling. For ODEBRAIN norm with $\tau$=3 gives the best

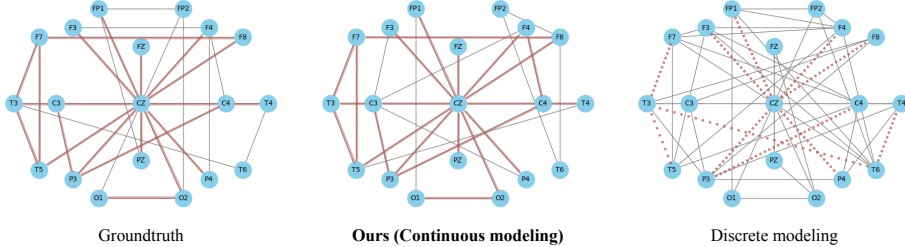

(a) Comparison among groundtruth, graph output of our continuous predictor, graph output of discrete predictor.



(b) Comparison of correlation scores between graph output of our continuous predictor, and graph output of discrete predictor.

Figure 9: Comparison of the predicted graph output between our continuous predictor and discrete predictor.

AUROC (0.881), whereas graphical lasso with $\tau$=3 achieves the highest Recall (0.613); the performance gap is small between $\tau$ and regularizers, demonstrating robust behavior to graph-construction choices.

Table 6 shows the effects of GNN backbones on TUSZ under 12s and 60s forecasting horizons. We find that the GNN choice has a nontrivial impact on continuous seizure forecasting. GRU-GCN yields the best overall performance, reaching 0.881 AUROC / 0.496 F1 at 12s and 0.828 AUROC / 0.430 F1 at 60s. This indicates that recurrent gating over graph messages better captures fast and non-stationary dynamics, especially for short-term prediction. DCRNN performs competitively but consistently below GRU-GCN (0.823/0.433 at 12s; 0.818/0.417 at 60s), suggesting diffusion-based spatiotemporal propagation is effective but less expressive without explicit gating. In contrast, EvolveGCN degrades substantially, particularly for

Table 6: Ablation of GNN options on TUSZ (12s and 60s seizure detection) (AUROC↑, F1↑) Bold = best.

| | Method | T(Sec.) | AUROC | F1 |
|---|---|---|---|---|
| ODE Temporal-spatial | EvolveGCN | 12 | 0.791±0.003 | 0.401±0.002 |
| | | 60 | 0.729±0.002 | 0.378±0.003 |
| | DCRNN | 12 | 0.823±0.005 | 0.433±0.005 |
| | | 60 | 0.818±0.004 | 0.417±0.007 |
| | GRU-GCN | 12 | **0.881±0.006** | **0.496±0.017** |
| | | 60 | **0.828±0.003** | **0.430±0.021** |

long-horizon forecasting (0.729 AUROC / 0.378 F1 at 60s), implying that merely evolving GCN parameters is insufficient under noisy epoch-wise correlation graphs. In general, these results demonstrate that continuous latent dynamics ODEBRAIN benefit the most from temporally gated graph modeling, and the superiority is consistent across horizons.

Table 7 illustrates the robustness of ODEBRAIN when 30% of EEG segments are randomly masked, comparing it with latent-ODE. When 30% segments are randomly masked, ODEBRAIN exhibits smaller AUROC drops from 0.881 to 0.845, and F1 from 0.496 to 0.464; exceeding the AUROC and F1 of latent-ODE by 0.124 and 0.067, respectively. This demonstrates that ODEBRAIN maintains stable vector fields and detection performance under incomplete observations by leveraging adaptive gating operations within the vector field and stochastic regularization to suppress irregular time step

Table 8: Ablation on TUSZ dataset for 12s seizure detection with different top-$\tau$ options. **Bold** and underline indicate best and second-best results.

| Top-$\tau$ | AUROC | Recall | F1 |
|---|---|---|---|
| 2 | $0.867 \pm 0.003$ | $0.575 \pm 0.003$ | $0.484 \pm 0.009$ |
| 3 | $\mathbf{0.881 \pm 0.006}$ | $\mathbf{0.605 \pm 0.003}$ | $\mathbf{0.496 \pm 0.017}$ |
| 7 | $0.870 \pm 0.004$ | $0.602 \pm 0.004$ | $0.488 \pm 0.013$ |
| 9 | $0.868 \pm 0.004$ | $0.589 \pm 0.004$ | $0.487 \pm 0.011$ |
| 11 | $0.866 \pm 0.004$ | $0.571 \pm 0.002$ | $0.491 \pm 0.003$ |
| 13 | $0.865 \pm 0.003$ | $0.562 \pm 0.004$ | $0.474 \pm 0.003$ |

jumps. The results indicate that ODEBRAIN achieves robustness to trajectory uncertainty under the effects of missing values, enhancing the capacity of ODE solvers.

Table 8 shows the effects of the sparsity level of the correlation graph, controlled by the Top-$\tau$ neighbors per node. Overall, AUROC remains stable performance across $\tau$ from 2 to 13 (0.865–0.881), indicating that ODEBRAIN is not overly sensitive to Top-$\tau$ options. $\tau = 3$ achieves the best AUROC (0.881) and F1 (0.496), while both too sparse ($\tau = 2$) and too dense graphs ($\tau \geq 9$) lead to slight degradation. When the values of $\tau$ is small, the graph becomes too sparse making the edge GRU forward stage affect the quality of the graph descriptor. As $\tau$ increases, edges become much denser and correlation-based connectivity con-

Table 7: Ablation of missing value (MV) on TUSZ (12s seizure detection) with AUROC↑, F1↑, and predicted missing graph structural similarity (Sim.)↑ (Bold = best).

| MV | Method | Sim. | AUROC | F1 |
|---|---|---|---|---|
| 0% | latent-ODE | 0.53 | $0.791 \pm 0.003$ | $0.401 \pm 0.002$ |
| | ODEBRAIN | **0.63** | $\mathbf{0.881 \pm 0.006}$ | $\mathbf{0.496 \pm 0.017}$ |
| 30% | latent-ODE | 0.41 | $0.721 \pm 0.004$ | $0.377 \pm 0.003$ |
| | ODEBRAIN | **0.55** | $\mathbf{0.845 \pm 0.002}$ | $\mathbf{0.464 \pm 0.007}$ |

tains propagated noise, which makes the edge GRU forward more over-smoothing and injects noise structure into the initial state $z_0$. The denser Top-$\tau$ reduces the robustness of the vector field $f_\theta$. Therefore, we adopt $\tau = 3$ as a good trade-off between predictive performance and robustness of ODE dynamics.

