# OpenReview forum: "ODEBrain: Continuous-Time EEG Graph for Modeling Dynamic Brain Networks"
_ICLR.cc/2026/Conference — ICLR 2026 Poster_

### Official Review · Reviewer_bZrF · 2025-10-29

**Soundness:** 2
**Presentation:** 2
**Contribution:** 2
**Rating:** 2
**Confidence:** 4

**Summary:**

This paper proposes ODEBRAIN, a neural ODE latent dynamic forecasting framework with integrating spatio-temporal features into spectral graph nodes. The continuous latent dynamics was modeled by a neural ODE. The model shows efficiency in forecasting EEG dynamics.

**Strengths:**

The paper extends graph-based EEG modeling into the continuous domain using Neural ODEs, with separating spatial and temporal encoders, the method combines structured connectivity with temporal uncertainty. The proposed method does show better performance than several baselines.

**Weaknesses:**

1. The objective function used to train the algorithm is not clearly defined.
2. The method should include comparisons with other continuous-time baselines, such as BrainODE, which is already cited in the related works. Such comparisons would help demonstrate the specific advantages of introducing the graph-based formulation.
3. While the proposed method introduces a graph structure, it might be the main distinction from existing ODE approaches, the paper does not clearly explain how and why incorporating the graph improves forecasting of temporal dynamics.
4. The data is processed by STFT, however, the details of STFT are not included in the paper, and how do different parameters (such as window, frequency bins, log scaling) influence the performance of the model?
5. The paper does not discuss limitations of the proposed method.

**Questions:**

1. In 4.1, how to decide top-tau?
2. Some typo: line 219, space before ‘Consequently’. Line 21 and line 23, ‘verifies’ to ‘verify’.

---

> ### Author Response · Authors · 2025-11-24
>
> We appreciate your time and the recognition of the strength of our work.
> Below, we address each weakness and question with additional analysis and clarifications that we will incorporate into the revised version.
> >Regarding W1, no clearly defined objective function.
>
>
> Thank you for your comments, and we have provided the objective function in Lines 295-303.
>
> >Regarding W2, no comparisons with other continuous-time baselines.
>
> Thank you for your comments, and we have added the continuous-time baselines in main results (Table 1 in revised manuscript).
> ### Table 1 Main results on TUSZ.
>
> | Model | Acc   | F1   | AUROC  |
> | ---------- | ----------------- | ----------------- | ----------------- |
> | latent-ODE       | 0.827±0.004    | 0.470±0.005     | 0.849±0.004 |
> | latent-ODE(RK4)       | 0.821±0.003    | 0.465±0.007     | 0.845±0.004 |
> | ODE-RNN       | 0.802±0.002    | 0.455±0.007     | 0.855±0.003 |
> | neural-SDE       | 0.857±0.002    | 0.467±0.003     | 0.851±0.002 |
> | Graph-ODE       | 0.849±0.002    | 0.475±0.003     | 0.841±0.003 |
> | Ours       | **0.877±0.002**    | **0.496±0.017**     | **0.881±0.003** |
> ### Table 1 Main results on TUAB.
>
> | Model | Acc   | F1   | AUROC  |
> | ---------- | ----------------- | ----------------- | ----------------- |
> | latent-ODE       | 0.749±0.003    | 0.745±0.002     | 0.829±0.004 |
> | latent-ODE(RK4)       | 0.746±0.002    | 0.739±0.002     | 0.823±0.003 |
> | ODE-RNN       | 0.751±0.003    | 0.744±0.004     | 0.838±0.005 |
> | neural-SDE       | 0.768±0.003    | 0.751±0.003     | 0.834±0.002 |
> | Graph-ODE       | 0.757±0.003    | 0.737±0.006     | 0.823±0.004 |
> | Ours       | **0.778±0.003**    | **0.774±0.005**     | **0.857±0.005** |
>
> >Regarding W3, no disscussion with other existing ODE approaches.
>
> Thank you for your comments, we disscused the existing ODE approaches in the related works and analysised the challenges in preliminary.
>
> The discussion of prior work on modeling EEG signals with controllable formulations and fractional dynamics provide important theoretical foundations for modeling brain dynamics [1,2,4,5] , nonlinear EEG analysis for brain activity mining with theortical ODE funtion [3], differentiable ODE approaches for the EEG downstream task [6,7]. [1] G. Gupta et al., "Re-thinking EEG-based non-invasive brain interfaces: Modeling and analysis." In 2018 ACM/IEEE 9th International Conference on Cyber-Physical Systems (ICCPS), pp. 275-286. IEEE, 2018. [2] V. Tzoumas et al., "Selecting sensors in biological fractional-order systems." IEEE Transactions on Control of Network Systems 5, no. 2 (2018): 709-721. [3] JPM. Pijn et al., "Nonlinear dynamics of epileptic seizures on basis of intracranial EEG recordings." Brain topography 9, no. 4 (1997): 249-270. [4] X. Lu, "Detection and classification of epileptic EEG signals by the methods of nonlinear dynamics." Chaos, Solitons & Fractals 151 (2021): 111032. [5] R. Martis et al. "Epileptic EEG classification using nonlinear parameters on different frequency bands." Journal of Mechanics in Medicine and Biology 15, no. 03 (2015): 1550040. [6] Y. Chen et al., "EEG emotion recognition based on ordinary differential equation graph convolutional networks and dynamic time wrapping." Applied Soft Computing 152 (2024): 111181. [7] Han K, Yang Y, Huang Z, et al. Brainode: Dynamic brain signal analysis via graph-aided neural ordinary differential equations[C]//2024 IEEE EMBS International Conference on Biomedical and Health Informatics (BHI). IEEE, 2024: 1-8.

---

> ### Author Response · Authors · 2025-11-24
>
> >Regarding W4, no discussion on the limitations of the proposed method.
>
> Thank you for your comments. We discussed the limitations in the revised manuscript in Lines 538-539.
>
> >Regarding Q1, no disscussion with top-$\tau$ setting.
>
> Thank you for your comments, and we have added a new table (Table 4 in the revised manuscript) to show sensitivity with different top-$\tau$, and correlations.
>
> ### Table 4 Ablation on different Top-$\tau$ and regularizer options.
>
> | Model | Regularizer   | Top-$\tau$   | AUROC  | Recall |
> | ---------- | ----------------- | ----------------- | ----------------- | ----------------- |
> | latent-ODE       | Shrinkage     | 3     | 0.833±0.032     |0.567±0.021
> |   |      | 7     | 0.829±0.039     |0.554$\pm$0.032
> |   | Graphical lasso   | 3 | 0.846±0.025 | 0.557±0.022
> |   |    | 7 |0.841±0.036| 0.531±0.031
> |   |  Norm  | 3     |0.849±0.004   |0.575±0.005
> |   |        | 7     | 0.838±0.034   |0.545±0.043
> | Ours       | Shrinkage    | 3     | 0.872±0.023    | 0.606±0.035
> |      |    | 7     | 0.868±0.034     |0.594±.043
> | | Graphical lasso     | 3     | 0.872±0.017    | \bf 0.613±0.033
> | |     | 7     | 0.874±0.029     |0.607±0.004
> | | Norm    | 3     | 0.881±0.006    |0.605±0.003
> | |    | 7     | 0870±0.004    |0.602±0.004
>
> >Regarding Q2, some typo issues.
>
> thank you, we have corrected typo errors.

---

> > ### Comment · Reviewer_bZrF · 2025-11-28
> >
> > I thank the authors for their detailed responses to my questions. In recognition of their efforts until now, I will raise my score, as the proposed model does outperform the existing baselines included in the paper. I am increasing my score to a borderline.
> >
> > However, I still have one remaining concern. Could the authors briefly discuss why their method achieves better performance? For instance, is the improvement primarily driven by the graph construction? or the top-tau strategy? or the reverse initial encoding? or the specific spatial and temporal decoding components? or all of above? A short discussion would greatly clarify the source of the model’s advantage.

---

> ### Author Response · Authors · 2025-11-28
>
> Dear Reviewer bZrF,
>
> We sincerely appreciate **your recognition and the raised score**. We are very glad to hear that our responses have addressed your concerns.
>
> Regarding the remaining concern, the performance improvement comes from several complementary design choices, summarized below.
>
> Conceptually, the major gain comes from explicitly modeling continuous dynamics over graph structures. By capturing the dynamic evolution of EEGs, the model can handle substantial noise, randomness, and fluctuations. Our comparison with the baseline w/o continuous dynamics (i.e., using only a temporal GNN backbone) clearly supports this observation.
> Methodologically, we focus on *how to obtain a high-quality initialization $z_0$* and *how to formulate a vector field $f_\theta$ that captures informative dynamics*:
> * Reverse initial encoding provides a high quality continuous representation $z_0$ that enables the model to unfold temporal information embedded in EEGs. This is achieved by the dual-encoder architecture, combining spectral graph features with stochastic temporal signals.
> * The temporal–spatial ODE solver $f_\theta$ incorporates the initialization into additive and gating operations, enabling adaptive emphasis on informative EEG connectivity patterns that encode richer dynamics (new Figure 3 in the revised manuscript).
> * The stochastic regularizer (Eq.4) mitigates the classical error accumulation problem of ODEs by modeling stochasticity in the EEG time domain.
> We have also added a new ablation table (Table 2 in the revised manuscript) to support the above points.
>
> | Method | T(s)   |  AUROC | F1 |
> | ---------- | ----------------- | ----------------- | ----------------- |
> | Ours     | 12     | **0.881±0.006**      | **0.496±0.017**     |
> |   |  60    | **0.828±0.003**   | **0.430±0.021**    |
> |  Ours w/o ODE | 12   | 0.816±0.002  | 0.416±0.005 |
> |   |   60   | 0.802±0.003    | 0.413±0.005     |
> | Ours w/o gate     | 12     | 0.867±0.004      | 0.488±0.007     |
> |   |  60    | 0.821±0.034   | 0.424±0.003  |
> | Ours w/o stochastic    | 12     | 0.848±0.017      | 0.462±0.013     |
> |   |  60    | 0.817±0.029   | 0.414±0.047    |
> | Ours w/ random     | 12     | 0.860±0.017      | 0.474±0.033    |
> |   |  60    | 0.819±0.0026   | 0.418±0.017    |
>
>
> We have summarized all of the above discussion in Appendix Discussion Section of the revised manuscript.
>
> We sincerely appreciate your time, consideration, and constructive discussion throughout the review process. We kindly hope the reviewer will reconsider the rating.
>
>
>
> Authors

---

### Official Review · Reviewer_xvZK · 2025-10-31

**Soundness:** 3
**Presentation:** 4
**Contribution:** 3
**Rating:** 6
**Confidence:** 4

**Summary:**

This well-written paper introduces ODEBRAIN, a continuous-time latent dynamics framework for multi-channel EEG that: (1) builds spectral EEG graphs (where nodes are channels; edges are top-k correlations in STFT space); (2) uses a dual encoder to initialize a Neural ODE (NODE) with a deterministic graph descriptor and a stochastic temporal descriptor; and (3) forecasts future graph node embeddings via a graph-prediction head with multi-step loss. The authors additionally visualize the learned vector field and argue it is clinically interpretable (e.g., “centers” during seizures). They claim the formulation is the first to explicitly cast EEG brain networks as a continuous-time dynamical system governed by a NODE and show gains on two real world data sets over discrete baselines.

**Strengths:**

The paper argues discretized, windowed EEG pipelines miss inherently continuous dynamics, motivating a NODE approach and posing concrete challenges (robust initialization; meaningful trajectory objectives).

The dual-encoder (graph plus stochastic temporal) initialization and a graph-forecasting head with multi-step loss are well aligned with continuous latent trajectory learning. The forward ODE is standard and the projection aims at future graph prediction, not just signal.

Visualizations of the learned vector field highlight identifiable structures during seizure vs. non-seizure (e.g., attractor-like “centers” only during seizures), which is a promising narrative for clinical insight.

On TUSZ/TUAB EEG data, ODEBRAIN outperforms CNN-LSTM, BIOT, EvolveGCN, DCRNN, and AMAG in AUROC/F1. Both single-step and multi-step settings are reported and ablations probe initialization choices, loss design, and horizon.

Data, preprocessing, solver tolerance, training hyperparameters, and an anonymous code link are provided to aid reproducibility.

**Weaknesses:**

Although the latent dynamics are continuous via NODE, inputs and supervision remain epoched STFT segments and edges are top-k correlations per epoch. The model forecasts at discrete horizons (1s/3s/11s), and training targets are per-epoch graphs.

Sensitivity analyses related to top-\tau sparsity and normalized correlations are not reported.

The baselines considered are primarily discrete TGNs/transformers. The related-work cites graph ODEs, but the empirical table omits them.

NODE solvers can be costly/unstable. You specify RK45 and tolerances, but there’s no wall-clock / NFEs / memory vs. discrete baselines, nor sensitivity to tolerances/horizons

**Questions:**

Please clarify in what sense the approach exceeds a finely sampled discrete model, beyond the solver’s internal substeps; e.g., can ODEBRAIN answer arbitrary-time queries between epochs evaluated against held-out high-rate labels?

Edges use normalized correlation with top-\taup sparsification. Correlation graphs are sensitive to noise and volume conduction; what happens with different \tau, similarity metrics, or regularizers (e.g., shrinkage/graphical lasso/lagged connectivity)?

Since the proposed benefit is about continuous-time superiority, why aren’t Latent ODE-style baselines (e.g., NODE on per-channel embeddings without graph, and graph-ODE methods) and irregular-sampling setups considered?

---

> ### Author Response · Authors · 2025-11-24
>
> We appreciate your time and the recognition of the strength of our work.
> Below we address each weakness and question with additional analysis and clarifications that we will incorporate into the revised version.
>
> >Regarding Q1, the approach exceeds a finely sampled discrete model, beyond the solver’s internal substeps; e.g., can ODEBRAIN answer arbitrary-time queries between epochs evaluated against held-out high-rate labels"
>
> Thank you for your discussion. Yes, our method provides continuous-time latent dynamics, enabling arbitrary-time queries beyond discrete sampling resolution.
>
> * We understand your concern: a discrete model with sufficiently high-rate labels may appear to approximate a continuous function, but doing so requires an intractably large number of steps and quadratic computational cost in practice. In contrast, our ODE-based formulation learns a continuous latent dynamics with only a few discretized steps, yet provides a fully continuous-time initialization.
> * Unlike a finely sampled discrete model, the trajectory between training timestamps is not obtained by interpolation but is governed by the learned ODE function. As such, ODEBRAIN can evaluate predictions at arbitrary intermediate times and can be validated against high-rate held-out labels. In the paper, we provide an irregular sampling ablation to show that our method can infer dynamics for unseen time index.
>
> ### Table 7 Missing value (MV) results on TUAB 12s seizure detection.
> | MV | Method   | Sim.   | AUROC  | Recall |
> | ---------- | ----------------- | ----------------- | ----------------- | ----------------- |
> | 0%       | latent-ODE    | 0.53     | 0.791±0.003     |0.401±0.002
> |        | Ours    | 0.63     | 0.881±0.006     |0.496±0.017
> | 30%       | latent-ODE    | 0.41     | 0.721±0.004     |0.377±0.003
> |        | Ours    | 0.55     | 0.845±0.002     |0.464±0.007
>
>
>
> >Regarding Q2, sensitivity analyses related to top-$\tau$ sparsity and regularizers.
>
> Thank you for your comments, and we have added a new table (Table 4 in revised manuscript) to show sensitivity with different top-$\tau$, correlations.
>
> ### Table 4 Ablation on Top-$\tau$ and different regularizer options.
>
> | Model | Regularizer   | Top-$\tau$   | AUROC  | Recall |
> | ---------- | ----------------- | ----------------- | ----------------- | ----------------- |
> | latent-ODE       | Shrinkage     | 3     | 0.833±0.032     |0.567±0.021
> |   |      | 7     | 0.829±0.039     |0.554$\pm$0.032
> |   | Graphical lasso   | 3 | 0.846±0.025 | 0.557±0.022
> |   |    | 7 |0.841±0.036| 0.531±0.031
> |   |  Norm  | 3     |0.849±0.004   |0.575±0.005
> |   |        | 7     | 0.838±0.034   |0.545±0.043
> | Ours       | Shrinkage    | 3     | 0.872±0.023    | 0.606±0.035
> |      |    | 7     | 0.868±0.034     |0.594±.043
> | | Graphical lasso     | 3     | 0.872±0.017    | \bf 0.613±0.033
> | |     | 7     | 0.874±0.029     |0.607±0.004
> | | Norm    | 3     | 0.881±0.006    |0.605±0.003
> | |    | 7     | 0870±0.004    |0.602±0.004

---

> > ### Author Response · Authors · 2025-11-24
> >
> > >Regarding Q3 \& W3, no disscussion with other existing ODE approaches, and irregular-sampling setups considered.
> >
> > Thank you for your comments, and we have added two new tables, one (Table 1 in revised manuscript) is for the ODE-related results in main results; one (Table 7 in appendix) is for the irregular-sampling setups to compare.
> > ### Table 1 Main results on TUSZ.
> >
> > | Model | Acc   | F1   | AUROC  |
> > | ---------- | ----------------- | ----------------- | ----------------- |
> > | latent-ODE       | 0.827±0.004    | 0.470±0.005     | 0.849±0.004 |
> > | latent-ODE(RK4)       | 0.821±0.003    | 0.465±0.007     | 0.845±0.004 |
> > | ODE-RNN       | 0.802±0.002    | 0.455±0.007     | 0.855±0.003 |
> > | neural-SDE       | 0.857±0.002    | 0.467±0.003     | 0.851±0.002 |
> > | Graph-ODE       | 0.849±0.002    | 0.475±0.003     | 0.841±0.003 |
> > | Ours       | **0.877±0.002**    | **0.496±0.017**     | **0.881±0.003** |
> > ### Table 1 Main results on TUAB.
> >
> > | Model | Acc   | F1   | AUROC  |
> > | ---------- | ----------------- | ----------------- | ----------------- |
> > | latent-ODE       | 0.749±0.003    | 0.745±0.002     | 0.829±0.004 |
> > | latent-ODE(RK4)       | 0.746±0.002    | 0.739±0.002     | 0.823±0.003 |
> > | ODE-RNN       | 0.751±0.003    | 0.744±0.004     | 0.838±0.005 |
> > | neural-SDE       | 0.768±0.003    | 0.751±0.003     | 0.834±0.002 |
> > | Graph-ODE       | 0.757±0.003    | 0.737±0.006     | 0.823±0.004 |
> > | Ours       | **0.778±0.003**    | **0.774±0.005**     | **0.857±0.005** |
> >
> > ### Table 7 Missing value (MV) results on TUAB 12s seizure detection.
> > | MV | Method   | Sim.   | AUROC  | Recall |
> > | ---------- | ----------------- | ----------------- | ----------------- | ----------------- |
> > | 0%       | latent-ODE    | 0.53     | 0.791±0.003     |0.401±0.002
> > |        | Ours    | 0.63     | 0.881±0.006     |0.496±0.017
> > | 30%       | latent-ODE    | 0.41     | 0.721±0.004     |0.377±0.003
> > |        | Ours    | 0.55     | 0.845±0.002     |0.464±0.007
> >
> > >Regarding W4, "NODE solvers can be costly/unstable. You specify RK45 and tolerances, but there’s no wall-clock / NFEs / memory vs. discrete baselines, nor sensitivity to tolerances/horizons."
> >
> > Thank you for your comments, and we have added two new tables, one (Table 3 in revised manuscript) is for computational cost, wall-clock within single test batch (256), and number of function evaluations (NFEs), one (Table 2 in revised manuscript) is for sensitvity to long horizons.
> >
> > ### Table 3 Computational cost
> >
> > | Model | Param.   | Wall-clock(s)   | NFEs  |
> > | ---------- | ----------------- | ----------------- | ----------------- |
> > | CNN-LSTM       | 5976K     | 0.586 ± 0.004     | -     |
> > | BIOT   | 3174K | 0.508 ± 0.003 | - |
> > | DCRNN       | 281K     | 0.418 ± 0.006     | -     |
> > | latent-ODE       | 386K     | 0.421 ± 0.002     | 102     |
> > | ODE-RNN       | 675K     | 0.601 ± 0.005     | 189     |
> > | neural-SDE       | 346K     | 0.482 ± 0.003     | 153     |
> > | ODEBRAIN (Ours)       | 459K     | 0.516 ± 0.002     | 164     |
> >
> > ### Table 2 Results on TUSZ (12s and 60s seizure detection).
> >
> > | Method | T(s)   |  AUROC | F1 |
> > | ---------- | ----------------- | ----------------- | ----------------- |
> > | BIOT       | 12     | 0.772±0.006      | 0.294±0.006     |
> > |   |  60    | 0.642±0.009     |   0.256±0.003   |
> > |  DCRNN | 12   | 0.816±0.002  | 0.416±0.005 |
> > |   |   60   | 0.802±0.003    | 0.413±0.005     |
> > |  latent-ODE | 12   | 0.791±0.004  | 0.385±0.005 |
> > |   |   60   | 0.745±0.004    | 0.331±0.007     |
> > | Ours     | 12     | **0.881±0.006**      | **0.496±0.017**     |
> > |   |  60    | **0.828±0.003**   | **0.430±0.021**    |

---

> > > ### Comment · Reviewer_xvZK · 2025-11-26
> > > **Response to revisions**
> > >
> > > We appreciate the authors additional experiments. Would it be possible to tune hyperparameter \tau over more than two values? We appreciate timing here is tight, but [3,7] is hardly a grid. Is the method too heavy computationally to readily run it over a more dense grid for \tau?

---

> ### Author Response · Authors · 2025-11-28
>
> Thank you for your response, and we have added a new table to show sensitivity with different top-$\tau$ here and in table 8, Appendix.
> Regarding the computation cost, the method remains feasible. Increasing top-$\tau$ from 3 to 13 adds approximately 10 additional channels, which increases the per-batch computation time by about 35%. This overhead only affects the ODE initialization in the forward graph-embedding computation. The cost does not come from the ODE solver itself; rather, a denser graph connectivity makes the GNN aggregation become slightly more expensive.
> Overall, the impact is moderate and does not significantly hinder scalability.
>
> | Top-$\tau$ | AUROC            | Recall         | F1             |
> | ---------- | ---------------- | -------------- | -------------- |
> | 2          | 0.867±0.003      | 0.575±0.003    | 0.484±0.009    |
> | 3 (Ours)   | 0.881±0.006      | 0.605±0.003    | 0.496±0.017    |
> | 7          | 0.870±0.004      | 0.602±0.004    | 0.488±0.013    |
> | 9          | 0.868±0.004      | 0.589±0.004    | 0.487±0.011    |
> | 11         | 0.866±0.004      | 0.571±0.002    | 0.491±0.003    |
> | 13         | 0.865±0.003      | 0.562±0.004    | 0.474±0.003    |
> | 15         | 0.864±0.003      | 0.561±0.003    | 0.472±0.005    |
>
>
> We sincerely appreciate your time, consideration, and constructive discussion throughout the review process. We kindly hope you further support our work.

---

### Official Review · Reviewer_GEpk · 2025-11-01

**Soundness:** 2
**Presentation:** 2
**Contribution:** 2
**Rating:** 4
**Confidence:** 5

**Summary:**

This paper introduces ODEBRAIN, a framework for modeling dynamic brain networks from multi-channel EEG. Unlike existing work modeling EEG dynamics in discrete-time manner, the method aims to learn the continuous-time representation of brain state evolution by leveraging Neural Ordinary Differential Equation (NODE). To address the challenges of applying NODEs to EEG, the authors propose a dual-encoder architecture to obtain robust initialization for ODE solver, combining spectral graph features with stochastic temporal signals. Then a novel objective function is proposed to capture underlying EEG dynamics. Experiments on two seizure detection datasets demonstrate model effectiveness.

**Strengths:**

- The paper addresses an important problem that of modeling brain network in continuous time (with NODE).
- The novel contributions mainly come from the proposed dual-encoder architecture and objective loss.
- The learned dynamic field demonstrate potential for qualitative analyses of brain networks.
- The proposed method shows some improvement, especially on TUSZ.

**Weaknesses:**

- Several Technical details of the proposed method are not clearly stated / explained. How do the the temporal descriptor \Psi and the objective \Omega are defined? How the pooling of latent continuous trajectory z_t for downstream task is conducted?
- The paragraph “RQ3 concerns consistency in the graphs…” seems confusing. From Fig.4 I cannot interpret the similarity scores or similarity matrices from discrete or continuous predictor, and other discussion. It seems the figure misaligned with the content.
- Hyperparameter selection is not discussed. Why the paper uses top 3 correlation neighbors for each node when constructing the graph? How about the effectiveness of different GNN architectures?
- The complexity analysis for the model is missing. How is the training/inference time of ODEBRAIN compared to other baselines.
- What is the motivation for objective loss to predict graph structure rather than EEG signal forecasting? The proposed loss shows effectiveness for the prediction task, how it would perform on the typical EEG signal forecasting?
- The discussion of prior work on modeling EEG signals, capturing their nonlinearity for a variety of brain activity mining [1][3][4][8][14], modeling and control for analyzing EEG signals in the context of brain machine interfaces and epileptic seizures [5][6][12][13][15], modeling latent variables in EEG [7][11] and other brain activity data [9][10], as well as related papers using graph ODE for EEG data [1] are all missing. This needs improvement in both discussing existing methods and comparing against these methods.
[1] JPM. Pijn et al., "Nonlinear dynamics of epileptic seizures on basis of intracranial EEG recordings." Brain topography 9, no. 4 (1997): 249-270.
[2] Y. Chen et al., "EEG emotion recognition based on ordinary differential equation graph convolutional networks and dynamic time wrapping." Applied Soft Computing 152 (2024): 111181.
[3] Y. Xue et al., "Minimum number of sensors to ensure observability of physiological systems: A case study." In 2016 54th Annual Allerton Conference on Communication, Control, and Computing (Allerton), pp. 1181-1188. IEEE, 2016.
[4] K. Lehnertz et al. "Seizure prediction by nonlinear EEG analysis." IEEE Engineering in Medicine and Biology Magazine 22, no. 1 (2003): 57-63.
[5] G. Gupta et al., "Re-thinking EEG-based non-invasive brain interfaces: Modeling and analysis." In 2018 ACM/IEEE 9th International Conference on Cyber-Physical Systems (ICCPS), pp. 275-286. IEEE, 2018.
[6] V. Tzoumas et al., "Selecting sensors in biological fractional-order systems." IEEE Transactions on Control of Network Systems 5, no. 2 (2018): 709-721.
[7] G. Gupta et al., "Learning latent fractional dynamics with unknown unknowns." In 2019 American Control Conference (ACC), pp. 217-222. IEEE, 2019.
[8] K. Lehnertz, "Epilepsy and nonlinear dynamics." Journal of biological physics 34, no. 3 (2008): 253-266.
[9] R. Yang et al., "Data-driven perception of neuron point process with unknown unknowns." In Proceedings of the 10th ACM/IEEE International Conference on Cyber-Physical Systems, pp. 259-269. 2019
[10] R. Yang et al., "Spiking dynamics of individual neurons reflect changes in the structure and function of neuronal networks." Nature Communications 16, no. 1 (2025): 6994
[11] G. Gupta et al., "Dealing with unknown unknowns: Identification and selection of minimal sensing for fractional dynamics with unknown inputs." In 2018 Annual American Control Conference (ACC), pp. 2814-2820. IEEE, 2018.
[12] X. Lu, "Detection and classification of epileptic EEG signals by the methods of nonlinear dynamics." Chaos, Solitons & Fractals 151 (2021): 111032.
[13] R. Martis et al. "Epileptic EEG classification using nonlinear parameters on different frequency bands." Journal of Mechanics in Medicine and Biology 15, no. 03 (2015): 1550040.
[14] M. Mercier et al. "The value of linear and non-linear quantitative EEG analysis in paediatric epilepsy surgery: a machine learning approach." Scientific reports 14, no. 1 (2024): 10887.
[15] G. Lepeu et al. "The critical dynamics of hippocampal seizures." Nature communications 15, no. 1 (2024): 6945.
Claiming that the novelty of this approach is that it captures the connectivity of spatio-temporal graphs to capture nonstationary changes is incorrect as can be seen from existing works that do just that.
- The paper states that the proposed approach “combines deterministic graph-based features with stochastic EEG representations to produce a robust initial state” but it is unclear how the stochastic EEG is accounted for.
- Can we always assume that the data supports that f_theta is continuous and differentiable function? Some nonlinearity of EEG may not support this assumption.
- The parameters N, d and T are barely mentioned in passing in section 4.2 but how they are selected in the case study and some guidelines to select them are missing.

**Questions:**

- Can we always assume that the data supports that f_theta is continuous and differentiable function? Some nonlinearity of EEG may not support this assumption.
- Several Technical details of the proposed method are not clearly stated / explained. How do the the temporal descriptor \Psi and the objective \Omega are defined? How the pooling of latent continuous trajectory z_t for downstream task is conducted?
- The paragraph “RQ3 concerns consistency in the graphs…” seems confusing. From Fig.4 I cannot interpret the similarity scores or similarity matrices from discrete or continuous predictor, and other discussion. It seems the figure misaligned with the content.
- Hyperparameter selection is not discussed. Why the paper uses top 3 correlation neighbors for each node when constructing the graph? How about the effectiveness of different GNN architectures?
- The complexity analysis for the model is missing. How is the training/inference time of ODEBRAIN compared to other baselines.
- What is the motivation for objective loss to predict graph structure rather than EEG signal forecasting? The proposed loss shows effectiveness for the prediction task, how it would perform on the typical EEG signal forecasting?
- The discussion of prior work on modeling EEG signals, capturing their nonlinearity for a variety of brain activity mining [1][3][4][8][14], modeling and control for analyzing EEG signals in the context of brain machine interfaces and epileptic seizures [5][6][12][13][15], modeling latent variables in EEG [7][11] and other brain activity data [9][10], as well as related papers using graph ODE for EEG data [1] are all missing. This needs improvement in both discussing existing methods and comparing against these methods.

**Details Of Ethics Concerns:**

No comments.

---

> ### Author Response · Authors · 2025-11-24
>
> We appreciate your time and the recognition of the strength of our work.
> Below we address each weakness and question with additional analysis and clarifications that we will incorporate into the revised version.
>
>
>
> >Regarding Q1 \& W8, can we always assume that the data supports that f_theta is continuous and differentiable function? Some nonlinearity of the EEG may not support this assumption.
>
> Yes, we can. Our BRAINODE evaluates the continuous dynamic field in latent space, using its evolutionary state for interpretability and downstream tasks; rather than attempting to describe the raw EEG is differentiable.
>
> > Regarding Q2 \& W1, details of temporal descriptor $\Psi$, the objective $\Omega$, and the pooling method of latent continuous trajectory $z_t$ for downstream task.
>
> Thank you for your comments, now we have provided details of temporal descriptor $\Psi$: we fellow the general CNN embedding method in [1].
> Details of $\Omega$: we project the continuous latent trajectories $\{\boldsymbol{z}(t)\}_{t=1}^{K}$ back to the future EEG node attributes with single-layer feedforward neural network.
>
> [1]A. M. Anwar and A. M. Eldeib, "EEG Signal Classification Using Convolutional Neural Networks on Combined Spatial and Temporal Dimensions for BCI Systems," 2020 42nd Annual International Conference of the IEEE Engineering in Medicine & Biology Society (EMBC)
>
> For compaing the effect of different pooling methods, we have added a new table (Table 5 in appendix), the results show the proposed method with the chosen max pooling achieved the best performance by perserving the dynamic diversity.
>
> ### Table 5 ablation of pooling choices on TUSZ.
>
> | Model | Acc   | F1   | AUROC  |
> | ---------- | ----------------- | ----------------- | ----------------- |
> | Max pooling       | **0.877±0.004**    | **0.496±0.017**     | **0.881±0.003** |
> | Mean pooling      | 0.842±0.002    | 0.385±0.005     | 0.827±0.004 |
> | Sum pooling       | 0.851±0.002    | 0.465±0.007     | 0.845±0.003 |
>
> ### Table 5 Ablation of pooling choices on TUAB.
>
> | Model | Acc   | F1   | AUROC  |
> | ---------- | ----------------- | ----------------- | ----------------- |
> | Max pooling       | **0.778±0.003**    | **0.774±0.005**     | **0.857±0.005** |
> | Mean pooling      | 0.735±0.002    | 0.635±0.002     | 0.827±0.004 |
> | Sum pooling      | 0.753±0.003    | 0.755±0.002    | 0.831±0.004 |

---

> > ### Author Response · Authors · 2025-11-24
> >
> > > Regarding Q3 \& W2, the paragraph “RQ3 concerns consistency in the graphs…” seems confusing. From Fig.4 I cannot interpret the similarity scores or similarity matrices from discrete or continuous predictor, and other discussion. It seems the figure misaligned with the content.
> >
> > Sorry for confusing with incorect figure 5 cition，now we have corrected the results in red marks of Line 472-478. Specifcally, Figure 6 shows the effectiveness of our objective $\Omega$ that helps predict dynamic graph structures. Our achieves higher similarity scores (0.53 $\rightarrow$ 0.63) in Figure 6 (a).
> >
> >
> > >Regarding Q4 \& W3, hyperparameter selection discussion of correlation neighbors for each node and the effectiveness of different GNN architectures.
> >
> > Thank you for your comments, and we have added two new tables, one (Table 4 in revised manuscript) is for top-$\tau$ correlation neighbors of graph construction, and one (Table 6 in appendix) is for the ablation of GNN architectures.
> >
> > ### Table 4 Ablation on Top-$\tau$ and different regularizer options.
> >
> > | Model | Regularizer   | Top-$\tau$   | AUROC  | Recall |
> > | ---------- | ----------------- | ----------------- | ----------------- | ----------------- |
> > | latent-ODE       | Shrinkage     | 3     | 0.833±0.032     |0.567±0.021
> > |   |      | 7     | 0.829±0.039     |0.554$\pm$0.032
> > |   | Graphical lasso   | 3 | 0.846±0.025 | 0.557±0.022
> > |   |    | 7 |0.841±0.036| 0.531±0.031
> > |   |  Norm  | 3     |0.849±0.004   |0.575±0.005
> > |   |        | 7     | 0.838±0.034   |0.545±0.043
> > | Ours       | Shrinkage    | 3     | 0.872±0.023    | 0.606±0.035
> > |      |    | 7     | 0.868±0.034     |0.594±.043
> > | | Graphical lasso     | 3     | 0.872±0.017    | \bf 0.613±0.033
> > | |     | 7     | 0.874±0.029     |0.607±0.004
> > | | Norm    | 3     | 0.881±0.006    |0.605±0.003
> > | |    | 7     | 0870±0.004    |0.602±0.004
> >
> > ### Table 6 Ablation of GNN options on TUSZ (12s and 60s seizure detection).
> >
> > | Method | T(s)   |  AUROC | F1 |
> > | ---------- | ----------------- | ----------------- | ----------------- |
> > | EvolveGCN       | 12     | 0.791±0.003      | 0.401±0.002     |
> > |   |  60    | 0.729±0.002     |   0.378±0.003   |
> > |  DCRNN | 12   | 0.823±0.005  | 0.433±0.005 |
> > |   |   60   | 0.818±0.004    | 0.417±0.007     |0.554$\pm$0.032
> > | GRU-GCN     | 12     | **0.881±0.006**      | **0.496±0.017**     |
> > |   |  60    | **0.828±0.003**   | **0.430±0.021**    |

---

> > > ### Author Response · Authors · 2025-11-24
> > >
> > > >Regarding Q5 \& W4, the complexity analysis for the model.
> > >
> > > Thank you for your comments, and we have added a new table (Table 3 in revised manuscript) to show computational cost, wall-clock within single test batch (256), and number of function evaluations (NFEs) for both descrete and continuous methods.
> > >
> > > ### Table 3 Computational cost
> > >
> > > | Model | Param.   | Wall-clock(s)   | NFEs  |
> > > | ---------- | ----------------- | ----------------- | ----------------- |
> > > | CNN-LSTM       | 5976K     | 0.586 ± 0.004     | -     |
> > > | BIOT   | 3174K | 0.508 ± 0.003 | - |
> > > | DCRNN       | 281K     | 0.418 ± 0.006     | -     |
> > > | latent-ODE       | 386K     | 0.421 ± 0.002     | 102     |
> > > | ODE-RNN       | 675K     | 0.601 ± 0.005     | 189     |
> > > | neural-SDE       | 346K     | 0.482 ± 0.003     | 153     |
> > > | ODEBRAIN (Ours)       | 459K     | 0.516 ± 0.002     | 164     |
> > >
> > >
> > >
> > > >Regarding Q6 \& W5, the motivation for objective loss to predict graph structure rather than EEG signal forecasting. The proposed loss shows effectiveness for the prediction task, how it would perform on the typical EEG signal forecasting?
> > >
> > > Raw EEG signals contain substantial noise, randomness, and non-deterministic fluctuations. Directly forecasting raw signals tends to amplify these stochastic components and does not necessarily reflect the underlying brain-state transitions we aim to model. Instead, our objective focuses on learning the evolution of graph states, which integrate information across multiple time steps and capture more stable, physiologically meaningful dynamics of brain connectivity.
> > >
> > > Addtionally, we have provided a new ablation study in Fig. 7b, when we substitute the objective with raw-data forecasting, the performance decreases. This supports our design choice that modeling graph-structure transitions is more effective and more aligned with the goal of capturing temporal evolution of brain network states。
> > >
> > >
> > > >RegardingQ7 \& W6, the discussion of prior work on modeling EEG signals.
> > >
> > > Thank you so much for your comments. We now have added the above literature in revised Related Work Section.
> > > Please kindly see it in Line 120-138.
> > >
> > >
> > > >Regarding W7, The paper states that the proposed approach “combines deterministic graph-based features with stochastic EEG representations to produce a robust initial state” but it is unclear how the stochastic EEG is accounted for.
> > >
> > >
> > > Since $z^{s}$ is obtained directly from raw EEG signals, it encodes the intrinsic randomness of brain activity as well as measurement noise and other non-deterministic fluctuations. These time-domain observations contain substantial stochastic components. In contrast, $z^{g}$ is constructed from FFT-based frequency representations, which are deterministic transformations of the signal and therefore filter out much of the randomness in the temporal domain. Thus, to explicitly model the uncertainty and variability inherent in raw EEG observations, we refer to $z^{s}$ as a stochastic temporal embedding.
> > >
> > >
> > > >Regarding W8, the parameters N, d and T are barely mentioned in passing in section 4.2 but how they are selected in the case study and some guidelines to select them are missing.
> > >
> > > * Actually, N is the number of channels of TUSZ/TUAB datasets, which is fixed parameter.
> > > * The selection of $d$ (FFT parameter) and $T$ (the number of time steps) refers to existing works of BIOT (NeurIPS 24), DCRNN (ICLR 2022), and EvoBrain (NeurIPS 25) for fair comparisons.
> > > * We additionally conduct an ablation study on T to examine the impact of different temporal window initializations, demonstrating how varying T affects performance.

---

> ### Author Response · Authors · 2025-11-28
>
> Dear Reviewer GEpk,
>
> We would like to once again express our gratitude for your insightful and valuable comments. As we approach the end of the discussion period, we hope that our responses have adequately addressed your concerns. If you have any questions or concerns would like to further discuss, feel free to share them with us. We thank you again for your time and hope will reconsider the rating.
>
> Authors

---

### Official Review · Reviewer_itCT · 2025-11-07

**Soundness:** 3
**Presentation:** 3
**Contribution:** 3
**Rating:** 6
**Confidence:** 3

**Summary:**

This paper presents ODEBRAIN, a novel continuous-time EEG graph framework built upon Neural Ordinary Differential Equations (NODEs) to model dynamic brain networks. To tackle key challenges in learning temporal brain dynamics, the study introduces three main contributions.

First, a dual-encoder architecture is proposed to effectively initialize NODEs: one encoder extracts deterministic frequency-domain features to represent brain connectivity, while the other processes raw EEG signals to preserve stochastic variability. Their integration provides robust spatiotemporal representations for initializing the ODE solver.

Second, a trajectory forecasting decoder is designed to reconstruct graph structures from NODE latent trajectories. By incorporating a multi-step forecasting loss, the model explicitly predicts the evolution of brain networks over time, enabling accurate and continuous trajectory modeling.

Third, the paper introduces a novel gradient field–based metric derived from NODEs to quantify the dynamics of EEG brain networks. A case study on seizure data demonstrates the clinical interpretability and practical value of this approach.

**Strengths:**

- The paper writing is generally good although some parts can be further improved.
- The idea of using ODE solver in forecasting and predicting graph structure are interesting.
- The experimental results are good.

**Weaknesses:**

- Some parts are not clear.
- No discussion of the computational cost.
- No discussion of the architecture of $f_\theta$.

**Questions:**

- What is GRU (Gated Recurrent Unit)?
- In the formula of $z^g$ in line 260, is it should be $z^g_i$?
- Can you explain more $z^s$? Why is it called stochastic temporal embedding?

---

> ### Author Response · Authors · 2025-11-24
>
> We appreciate your time and the recognition of the strength of our work.
> Below we address each weakness and question with additional analysis and clarifications that we will incorporate into the revised version.
>
> >Regarding no discussion of the computational cost.
>
> Thank you for your comments, and we have added a new table (Table 3 in revised manuscript) to show computational cost using total parameters, wall-clock within single test batch (256), and number of function evaluations (NFEs) for both descrete and continuous methods.
>
> ### Table 3 Computational cost
>
> | Model | Param.   | Wall-clock(s)   | NFEs  |
> | ---------- | ----------------- | ----------------- | ----------------- |
> | CNN-LSTM       | 5976K     | 0.586 ± 0.004     | -     |
> | BIOT   | 3174K | 0.508 ± 0.003 | - |
> | DCRNN       | 281K     | 0.418 ± 0.006     | -     |
> | latent-ODE       | 386K     | 0.421 ± 0.002     | 102     |
> | ODE-RNN       | 675K     | 0.601 ± 0.005     | 189     |
> | neural-SDE       | 346K     | 0.482 ± 0.003     | 153     |
> | ODEBRAIN (Ours)       | 459K     | 0.516 ± 0.002     | 164     |
>
>
> >Regarding no discussion of the architecture of $f_{\theta}$.
>
> We have provided more discussion of designs of $f_{\theta}$ in new figure 3 in manuscript. Specifically, we provide detailed temporal-spatial ODE solving to incorporate initial state $z_{0}$ for additive and gate operations. In addition, we further introduce an adaptive decay component conditioned on the stochastic temporal state $z^{s}$, to adjust the vector field $f_{\theta}$, accounting for the complexity and dynamic nature of the brain as a system.
>
>
> > Regarding GRU (Gated Recurrent Unit)"
>
> Yes, it is Gated Recurrent Unit.
>
>
> > Regarding  the formula of $z^{g}$ in line 260?"
>
> Thank you for discussion, and $z^{g}$ is correct as it is the intial states for all time steps, so there is no lower case $i$.
>
>
>
> > Regarding $z^{s}$ ? Why is it called stochastic temporal embedding?"
>
> Since $z^{s}$ is obtained directly from raw EEG signals, it encodes the intrinsic randomness of brain activity as well as measurement noise and other non-deterministic fluctuations. These time-domain observations contain substantial stochastic components. In contrast, $z^{g}$ is constructed from FFT-based frequency representations, which are deterministic transformations of the signal and therefore filter out much of the randomness in the temporal domain. Thus, to explicitly model the uncertainty and variability inherent in raw EEG observations, we refer to $z^{s}$ as a stochastic temporal embedding.

---

### Author Response · Authors · 2025-11-24
**Global response to all reviewers**

We thank all reviewers for their careful reading and helpful feedback. We have modified our paper according to the provided comments. Below is a list of the changes we made:

* (Reviewer itCT) new Table 3 that compares the computational cost of different methods.
* (Reviewer GEpk) new Table 5 in Appendix B that shows the performance across different pooling choices.
* (Reviewer GEpk) new Table 4 that compares the sensitivity with the top-correlation neighbors of graph construction.
* (Reviewer GEpk) new Table 6 in Appendix B that compares performance across GNN architectures.
* (Reviewer xvZK）add new main results in Table 1 that discuss with other existing ODE approaches.
* (Reviewer xvZK) new Table 7 in Appendix B that compares irregular-sampling setups.
* (Reviewer xvZK）new Table 4 that shows the sensitivity of both Top-k and regularizer options.
* (Reviewer bZrF）discuss the objective function in Lines 295-303.
* (Reviewer bZrF）add new main results in Table 1 that discuss with other existing ODE approaches.
* (Reviewer bZrF）typos fixed

---

### Comment · Area_Chair_rWyc · 2025-11-26
**Reminder to Engage!**

Dear Reviewers,

We are one week away from the end of the discussion period and the review responses have been posted. Please read the response and check if the authors have addressed your concerns. Also please acknowledge the review by responding and stating how the response (and updated manuscript if provided) does or does not change your evaluation of the work. Earlier responses allow for meaningful engagement and potential for further clarification.

-Area Chair

---

### Author Response · Authors · 2025-11-29
**Paper and Rebuttal Summary**

Dear Reviewers and ACs,

We sincerely thank all reviewers for their time, constructive feedback, and highly engaging discussion throughout the rebuttal period. We are grateful for the detailed examination of our work and for the many positive assessments we received regarding its *motivation* (itCT, GEpk, xvZK), *continuous-time level brain dynamics* (itCT, GEpk, xvZK, bZrF), *methodology (dual encoder and ODE-solver)* (itCT, GEpk, xvZK), *good performance and clinical interpretability* (itCT, GEpk, xvZK, bZrF), and *well-presentation* (itCT, xvZK).

We particularly appreciate the reviewer's in-depth follow-ups during the discussion phase, which greatly helped us strengthen the manuscript. We have incorporated all feasible suggestions and clarified every point raised. Below we summarize the key revisions made in response to reviewers' comments:
* Following the suggestions of GEpk,xvZK, and bZrF, we added **four ODE-related baselines** (neural ODE, ODE-RNN, neural SDE, neural GDE) (Table 1).
* Following the suggestions of itCT, GEpk, and xvZK, we conducted a **computational cost analysis**, including wall-clock time and the number of function evaluations (NFEs) for both discrete and continuous methods (Table 3).
* We added additional **methodological reflections**:
    * Following itCT, we added a more detailed explanation of the temporal-spatial ODE solving (line 257-269).
    * Following itCT, we added a new Figure 3 to illustrate the process of temporal-spatial ODE solving (P5).
    * Following GEpk, we clarified why the stochastics of raw EEGs contribute as an adaptive decay component during the ODE solving (line 275-282).
    * Following GEpk, we clarified that the effectiveness of objective loss to predict graph structure rather than EEG signal forecasting.
    * Following GEpk, we correct the Figure 6 citation (line 472).
    * Following GEpk, we added citations on the nonlinearity of brain activity, modeling latent variables in EEG, and modeling for analyzing EEG signals with theoretical ODEs in the *Related work*.
    * Following bZrf, we clarified the objective function (line 295-303).
    * Following bZrf, we clarified the existing ODE approaches in the related works and the challenges in the preliminary.
    * Following bZrf, we clarified that the model's advantage comes from several complementary design choices (P15).
* We added additional **ablation studies**:
    * Following GEpk, we added an ablation on pooling choices (Table 5 in Appendix B).
    * Following GEpk, we added an ablation on GNN architectures (Table 6 in Appendix B).
    * Following GEpk, we added an ablation on top-correlation neighbors of graph construction (top-tau)(Table 4).
    * Following xvZK, we added an ablation on regularizer options (Table 4).
    * Following xvZK, we added an ablation on dense top-$\tau$ grid (Table 8 in Appendix B).
    * Following bZrF, we added the discussion on top-$\tau$ (Table 4)

* We provided additional **robustness evaluations**:
    * Following xvZK, we added robustness analyses under long horizons (Table 2)
    * Following xvZK, we added irregular sampling (Table 7 in Appendix B).

During the rebuttal period, most reviewers offered positive feedback. In particular, **reviewer bZrF recognized our contributions and raised the score from 2 to 6**, and reviewer xvZK acknowledged the value of our additional experiments. For confirmation, **please kindly refer to the end of each reviewer’s response section.**

Thank you again for your time and consideration.

Authors

---

### Meta-Review · Area_Chair_yh6Z · 2026-01-05

**Summary:**

The main concerns of the Reviewers were: 1) missing/incomplete related work section, 2) missing baselines from the NODE community, 3) missing insights into computational cost, 4) more clarifications on the proposed model. This has led to an initial rating of 6,6,4,2. Based on the rebuttal, I believe that all major concerns have been addressed. I thus recommend acceptance.

**Reviewer Concerns:**

Reviewer itCT: only minor concerns on computational cost and architectural details. The rebuttal has addressed these.

Reviewer GEpk: Missing discussion on related and prior work has been provided in rebuttal. More details on computational cost, hyperparameter selection, and architectural details have been added. Most concerns have thus been adressed, and I believe that the reviewer would have raised to a 6.

Reviewer xvZK: sensitivity analysis has been provided in the rebuttal. Laten-ODE baseline have been added as well. Thus I believe the reviewer would have remained with their positive assessment.

Reviewer bZrF: All concerns were addressed and reviewer has raised score to 6 during discussion.

**Reviewer Scores:**

See above.

---

### Decision · Program_Chairs · 2026-01-26

Accept (Poster)